# How does Bayesian Sampling help Membership Inference Attacks?

Zhenlong Liu [1 2]   Wenyu Jiang [3]   Feng Zhou [4]   Hongxin Wei [1]

## Abstract

Membership Inference Attacks (MIAs) aim to estimate whether a specific data point was used in the training of a given model. Existing state-of-the-art attacks typically rely on training multiple reference models to approximate the conditional score distribution for individual data points, which leads to significant computational overhead and limits their practical applicability. In this work, we propose a novel approach – Bayesian Membership Inference Attack (BMIA), which performs conditional attack through Bayesian sampling. Specifically, we apply Laplace approximation to a single reference model to obtain a posterior over model parameters, enabling direct estimation of the conditional score distribution. Theoretically, we demonstrate that Bayesian sampling reduces intra-model variance, thereby improving attack power. This insight naturally motivates the multi-reference variant that further enhances performance when additional reference models are available. Extensive experiments across image, text, and tabular datasets indicate that our method achieves state-of-the-art performance in both effectiveness and efficiency. Our implementation code is available at https://github.com/zhenlong-liu/BMIA.

## 1. Introduction

Deep neural networks (DNNs) have achieved remarkable success in recent years, largely due to their high capacity stemming from extensive parameterization. This capacity enables them to learn complex patterns but also makes them vulnerable to memorizing training data, raising concerns about potential information leakage. One of the most widely used methods for quantifying data leakage is Membership Inference Attacks (MIAs), which aim to determine whether a specific example was part of a target model's training data. The success of MIAs serves as an indicator of privacy vulnerabilities (Jagielski et al., 2020b; Nasr et al., 2023), as it directly analyzes the differences in a model's behavior between its training data and unseen examples.

Some attackers utilize score functions for MIAs, such as loss (Yeom et al., 2018) and modified entropy (Song & Mittal, 2021). These score-based methods typically determine a global threshold by estimating the target model's marginal score distribution over the reference dataset, without tailoring inference to individual data points. To achieve conditional thresholds, previous works (Carlini et al., 2022; Watson et al., 2022; Ye et al., 2022) train multiple reference models to approximate the conditional score distribution for specific data points. However, those methods required training multiple models, suffering from significant computational overhead. This motivates us to develop an efficient conditional attack using a single reference model.

In this work, we propose a novel and cost-efficient membership inference attack – Bayesian Membership Inference Attack (**BMIA**), which estimates the conditional distribution using Bayesian sampling. In particular, we train a single reference model and then apply the Laplace approximation to estimate the posterior distribution of model parameters. This posterior allows us to directly obtain the predictive distribution of a given instance's outputs, thereby providing the conditional score distribution used for the conditional attack. We further provide a theoretical analysis showing that Bayesian sampling primarily reduces intra-model variance, which leads to improved statistical power. This analysis naturally extends to a multi-reference variant, termed Multi-Reference BMIA (MR-BMIA), which further improves the TPR when multiple reference models are available. Thus, BMIA provides a more precise estimation of uncertainty, resulting in an effective and efficient attack.

To verify the effectiveness of BMIA, we conduct extensive experiments on two tabular datasets, three image datasets, and five text datasets. Our method achieves a higher TPR at low FPR (e.g., $0.1\%$) than prior conditional attack methods with much higher efficiency. For example, our method

---

[1]Department of Statistics and Data Science, Southern University of Science and Technology [2]Shanghai Innovation Institute [3]School of Computer Science, Nanjing University [4]Center for Applied Statistics and School of Statistics, Renmin University of China . Correspondence to: Hongxin Wei <weihx@sustech.edu.cn>.

*Proceedings of the 43rd International Conference on Machine Learning*, Seoul, South Korea. PMLR 306, 2026. Copyright 2026 by the author(s).

achieves a true positive rate of 35.75% at a false positive rate of 1%, a 54% improvement over the prior state-of-the-art method (Carlini et al., 2022) on CIFAR-100, with just 1/8 computation cost. Moreover, we validate that our method can further improve attack performance when multiple reference models are available.

Our contributions are summarized as follows:

1. We introduce BMIA, a cost-effective and robust method for membership inference attacks. The core idea is to employ the Laplace approximation to derive the predictive score distribution, which requires training only a single reference model.

2. We theoretically establish that Bayesian sampling reduces intra-model variance, thereby provably increasing the attack power. This analysis naturally extends to a multi-reference variant (MR-BMIA).

3. We perform extensive experiments to illustrate that BMIA can achieve state-of-the-art performance with lower computational cost. These experiments are conducted on image, tabular, and text datasets, using a wide range of model architectures.

## 2. Background

**Setup.** In this paper, we study the problem of membership inference attacks in supervised learning. Let $z := (x, y) \in (\mathbb{R}^I \times \mathbb{R}^O)$ be an example containing an instance $x$ and a real-valued label $y$. Given a training dataset $\mathcal{S} = \{(x_i, y_i)\}_{i=1}^N$ *i.i.d.* sampled from the data distribution $\pi$, the weights $w \in \mathbb{R}^d$ of a machine learning model $f : \mathbb{R}^I \to \mathbb{R}^O$ are trained to minimize the empirical risk

$$\hat{w} = \arg\min_w [\sum_{i=1}^N \ell(f(x_i; w), y_i) + R(w)], \quad (1)$$

where $\ell$ is the loss function and $R : \mathbb{R}^d \to \mathbb{R}$ is the regularization function. For abbreviation, we denote the training algorithm as $\mathcal{T} : \mathcal{S} \to \hat{w}$.

**Bayesian Neural Networks (BNNs).** Instead of learning a set of fixed weights $\hat{w}$ like standard neural networks, BNNs treat weights as random variables sampled from a posterior distribution $p(w|\mathcal{D})$. From a Bayesian perspective, the standard training process (minimizing a regularized loss) can be interpreted as computing the Maximum A Posteriori (MAP) estimate, which identifies only the single most probable parameter configuration. However, relying solely on this point estimate fails to capture posterior uncertainty. In contrast, BNN inference leverages the full posterior distribution $p(w|\mathcal{D})$. Predictions are derived by marginalizing over the weights, accounting for all possible weight configurations and their associated probabilities:

$$p(y|x, \mathcal{D}) = \int p(y|x, w)p(w|\mathcal{D}) \, dw. \quad (2)$$

Since this integral is intractable for neural networks, it is typically approximated via Monte Carlo sampling by drawing multiple weight samples from the posterior, yielding a predictive distribution rather than a single point estimate.

**Membership Inference Attacks.** Given a trained model $f$ learned from a training dataset $\mathcal{S}$, membership inference attacks (MIAs) aim to determine whether a given data point $(x, y)$ was included in $\mathcal{S}$ (Shokri et al., 2017; Yeom et al., 2018; Salem et al., 2019; Zarifzadeh et al., 2024). We consider the standard inference setting for MIAs on a fixed target model, where the attacker is assumed to know the model architecture and the underlying data distribution. In this setting, a challenger samples a data point $(x, y)$ together with a membership label $m \in \{0, 1\}$, indicating whether $(x, y)$ is drawn from the training dataset $\mathcal{S}$ ($m = 1$) or from the data distribution $\pi$ ($m = 0$). The attacker queries the model predictions $f(x; w)$ and outputs a prediction $\mathcal{A}(x, y) \in \{0, 1\}$ for the membership label $m$. This inference task can also be formulated as a hypothesis-testing problem:

$$H_0 : (x, y) \in \pi \quad \text{vs.} \quad H_1 : (x, y) \in \mathcal{S} \quad (3)$$

In this framework, the attack can fix the probability of a Type I error, $\Pr(\text{reject } H_0 \mid m = 0)$, at a predetermined level $\alpha$ (i.e., the false positive rate) and then determine the corresponding threshold. A test that minimizes the Type II error, $\beta = \Pr(\text{accept } H_0 \mid m = 1)$, is considered more powerful. Note that the $1 - \beta$ is just the true positive rate (TPR), so the evaluation metric for MIAs is commonly the true positive rate at a low false positive rate.

To achieve the provable inference, the score-based method is the dominant method. As categorized by Ye et al. (Ye et al., 2022), the baseline method sets the threshold $\tau_\alpha$ to a constant value (Yeom et al., 2018). This approach, also referred to as the *marginal attack* (Bertran et al., 2024), is defined as:

$$\mathcal{A}(x, y; \alpha) = \mathbb{I}(s(x, y; \hat{w}) \geqslant \tau_\alpha), \quad (4)$$

where $\mathbb{I}$ is the indicator function, $s : \pi \to \mathbb{R}$ is a score function, e.g., confidence, loss or hinge score, $\tau_\alpha$ is the threshold satisfying:

$$\Pr(Q \geqslant \tau_\alpha) = \alpha, \quad (5)$$

where $Q$ denotes the random variable for $s(x, y; w)$ for $(x, y) \in \pi$, meaning the score distribution for the sample from the non-member set.

Though the marginal attack in the form of Equation (4) (Yeom et al., 2018; Salem et al., 2019; Song & Mittal, 2021) is widely used and effective, the constant threshold $\tau$ does not reflect the per-example hardness (Carlini et al., 2022). For instance, a hard example may consistently exhibit lower confidence unless it has been explicitly trained, leading to a constant threshold that more frequently identifies it as a non-member. In this way, the score $s(x, y; w)$ also does not reflect the privacy vulnerability of a sample, which means it cannot be viewed as a privacy metric (Aerni et al., 2024). Hence, the previous literature (Sablayrolles et al., 2019; Ye et al., 2022) considers the *conditional attack*:

$$\mathcal{A}(x, y, \alpha) = \mathbb{I}\left(s(x, y; \hat{w}) \geqslant \tau_\alpha(x, y)\right), \quad (6)$$

where $\tau_\alpha(x, y)$ is determined such that

$$\Pr(Q \geqslant \tau_\alpha | x, y) = \alpha \quad (7)$$

To determine the instance-wise threshold, prior works Attack-R (Ye et al., 2022), LiRA (Carlini et al., 2022) train multiple reference models (also referred to as shadow models), estimating the threshold through the conditional distribution. While effective, MIAs that rely on numerous references come with a significant computational cost. Such a requirement is nearly prohibitive for large-scale models due to the need to repeatedly train hundreds of full models, which severely limits the practicality of these attacks in real-world auditing scenarios. This motivates us to develop more efficient conditional MIAs that achieve comparable effectiveness using only one reference model.

## 3. Our Proposed Method

In this section, we introduce Bayesian Membership Inference Attack (**BMIA**), a novel attack that efficiently captures both aleatoric and epistemic uncertainty. This approach leverages Bayesian inference on a single reference model to estimate the conditional score distribution, which is then utilized to construct test statistics for inference.

The core principle of BMIA is to obtain the posterior distribution $p(w|\mathcal{D})$ from a standard reference model using the Laplace Approximation (LA) (Hahn, 2005; Daxberger et al., 2021). This posterior enables us to derive the predictive score distribution for a query point under the null hypothesis ($H_0$) of non-membership. Such an approach directly yields the conditional score distribution essential for instance-wise attacks, avoiding the computational expense of training multiple reference models and the limitations of quantile regression in fully capturing score uncertainty.

Operationally, a reference model is first trained on a dataset $\mathcal{D}$ (disjoint from member data) to obtain parameters $\hat{w}_1$. LA is then applied to approximate its weight posterior:

$$p(w|\mathcal{D}) \approx \mathcal{N}(w; \hat{w}_1, \Sigma), \quad (8)$$

---

**Algorithm 1 Bayesian Membership Inference Attack**

1: **Input:** Target model $w_0$, test instance $z^* = (x^*, y^*)$, and significance level $\alpha$, sample numbers $M$.
2: Randomly sample reference dataset $\mathcal{D} \sim \pi$
3: Compute target model score $s_0 \leftarrow s(x^*, y^*; w_0)$
4: Train a reference model $\hat{w}_1 \leftarrow \mathcal{T}(\mathcal{D})$
5: Obtain posterior distribution $\mathcal{N}(w; \hat{w}_1, \Sigma)$ via LA
6: **for** $i = 1$ to $M$ **do**
7:     Sample $\tilde{w}_i \sim \mathcal{N}(w; \hat{w}_1, \Sigma)$
8:     $s_i \leftarrow s(x^*, y^*; \tilde{w}_i)$
9:     $d_i \leftarrow s_0 - s_i$
10: **end for**
11: $\bar{d} \leftarrow \frac{1}{M} \sum_{i=1}^{M} d_i$
12: $\hat{\sigma} \leftarrow \sqrt{\frac{1}{M-1} \sum_{i=1}^{M} (d_i - \bar{d})^2}$
13: Compute test statistic $t \leftarrow \frac{\bar{d}}{\hat{\sigma}\sqrt{1+1/M}}$.
14: Compute p-value $p \leftarrow 1 - F_t(t; M - 1)$, where $F_t$ is the CDF of the $t$-distribution.
15: **Return** $m = 1$ if $p < \alpha$, and $m = 0$ otherwise.

---

where $\Sigma := \left(-\nabla_w^2 \mathcal{L}(\mathcal{D}; w)\big|_{w=\hat{w}}\right)^{-1}$

Given a target model with parameters $\hat{w}_0$ and a test data point $z^* := (x^*, y^*)$, multiple weight samples $\tilde{w}_i$ are drawn from the approximated posterior $p(w|\mathcal{D})$. For each sample $\tilde{w}_i$, a score $s_i = s(z^*; \tilde{w}_i)$ is computed. The resulting set of scores $\{s_i\}$ is then used to construct test statistic.

Recalling the attack defined in Equation (6), a higher score indicates a higher likelihood of membership. Therefore, we define a calibrated score $d_i = s_0 - s_i$ as the proxy for hypothesis testing, where $s_0 := s(z^*; \hat{w}_0)$ is the score obtained from the target model. Specifically, we adopt a one-sided, one-sample t-test, simplifying the hypothesis testing problem defined in Equation (3) as:

$$H_0 : \mathbb{E}[d_i] = 0 \quad \text{vs.} \quad H_1 : \mathbb{E}[d_i] > 0. \quad (9)$$

In this framework, $s_0$ is treated as a random variable. Under the null hypothesis that $z^*$ is a non-member, we assume $\text{Var}(s_0) = \text{Var}(s_i) = \sigma^2$. Consequently, the variance of $\bar{d} = \frac{1}{M} \sum_{i=1}^{M} (s_0 - s_i) = s_0 - \frac{1}{M} \sum_{i=1}^{M} s_i$ is derived as

$$\text{Var}(\bar{d}) = (1 + \frac{1}{M})\sigma^2. \quad (10)$$

Since the population variance $\sigma^2$ is unknown, we estimate it using the sample variance of the scores, $\hat{\sigma}^2$. The resulting test statistic $t = \bar{d}/(\hat{\sigma}\sqrt{1 + 1/M})$ follows a Student's $t$-distribution with $M - 1$ degrees of freedom. The complete procedure of our proposed BMIA is summarized in Algorithm 1.

In practice, we adopt the hinge score $s_{\text{hinge}}(x, y) = f(x)_y - \max_{y' \neq y} f(x)_{y'}$ as our certainty score function, where $f(x)$

is the logits vector. This score was introduced by Carlini et al. (2021) as a stable approximation for $\ln(\frac{p_y}{1-p_y})$, where $p_y$ denotes the confidence for the true label. Since it has been empirically shown to be approximately normally distributed, previous works (Carlini et al., 2022; Bertran et al., 2024) estimate the conditional score distribution based on this parametric assumption. Within our Bayesian inference framework, we assume that the model outputs follow a Gaussian distribution, leading to the hinge score also following a Gaussian distribution. Thus, our attack satisfies the Gaussian assumption underlying the Student's t-distribution. We provide empirical diagnostics for this approximation and discuss calibration under non-Gaussian scores in Appendix F.1.

### 3.1. Bayesian sampling helps reduce the variance

While our method leverages Bayesian sampling for conditional attacks, we first address a fundamental gap in the literature: the lack of a formal explanation for why conditional attacks outperform marginal ones. In Appendix B, we provide a theoretical analysis proving that conditional attacks yield a higher TPR by exploiting the distinctness of conditional score distributions. With this justification established, we next detail how Bayesian sampling further enhances performance via variance reduction.

Given a fixed sample $z$ and $K$ reference datasets $\{\mathcal{D}_k\}_{k=1}^K$ sampled from the distribution $\pi$, we consider the corresponding reference models. For each dataset $k$, the model parameters $\{w_{k,m}\}$ are sampled from the posterior distribution $p(w|\mathcal{D}_k)$. Let $s$ be a score sample drawn from the aggregate distribution of scores $\{s_{k,m}\}$, where $s_{k,m} := s(z; w_{k,m})$. By applying the Law of Total Variance, we can decompose the total variance of the score $s$ into two components:

$$\text{Var}(s) = \underbrace{\mathbb{E}[\text{Var}(s|\mathcal{D}_k)]}_{\sigma_{\text{intra}}^2} + \underbrace{\text{Var}[\mathbb{E}(s|\mathcal{D}_k)]}_{\sigma_{\text{inter}}^2}. \quad (11)$$

The first component $\sigma_{\text{intra}}^2$ (*intra-model variance*), arises from the uncertainty in model parameters $w$ given a fixed training dataset $\mathcal{D}$ (representing Bayesian posterior uncertainty). The second component $\sigma_{\text{inter}}^2$ (*inter-model variance*), explicitly captures the variability introduced by the data subsampling process. It quantifies how the expected score varies across reference models trained on different datasets $\mathcal{D}_k$. Then we establish the following proposition:

**Proposition 3.1.** *Let $s_0 := s(z; w_0)$ denote the score obtained from target model, $\bar{s} = \frac{1}{K}\sum_{k=1}^K \frac{1}{M}\sum_{m=1}^M s_{k,m}$. Under the $H_0$ hypothesis, the variance of $s_0 - \bar{s}$ is:*

$$\text{Var}(s_0 - \bar{s}) = (1 + \frac{1}{K})\sigma_{inter}^2 + (1 + \frac{1}{KM})\sigma_{intra}^2. \quad (12)$$

---

**Algorithm 2 Multi-Reference Bayesian Membership Inference Attack**

1: **Input:** Target model $w_0$, test instance $z^* = (x^*, y^*)$, significance level $\alpha$, number of reference models $K$, posterior samples per model $M$.
2: Compute target model score $s_0 \leftarrow s(x^*, y^*; w_0)$
3: **for** $k = 1$ to $K$ **do**
4:      Randomly sample reference dataset $\mathcal{D}_k \sim \pi$
5:      Train reference model $\hat{w}_k \leftarrow \mathcal{T}(\mathcal{D}_k)$
6:      Obtain posterior distribution $\mathcal{N}(w; \hat{w}_k, \Sigma_k)$ via LA
7:      **for** $i = 1$ to $M$ **do**
8:          Sample $\tilde{w}_{k,i} \sim \mathcal{N}(w; \hat{w}_k, \Sigma_k)$
9:          Compute score $s_{k,i} \leftarrow s(x^*, y^*; \tilde{w}_{k,i})$
10:      **end for**
11:      Compute per-model mean $\bar{s}_k \leftarrow \frac{1}{M}\sum_{i=1}^M s_{k,i}$
12: **end for**
13: $\bar{s} \leftarrow \frac{1}{K}\sum_{k=1}^K \bar{s}_k$
14: $\hat{\sigma}_{\text{intra}}^2 \leftarrow \frac{1}{K(M-1)}\sum_{k=1}^K\sum_{i=1}^M (s_{k,i} - \bar{s}_k)^2$
15: $\hat{\sigma}_{\text{inter}}^2 \leftarrow \frac{1}{K-1}\sum_{k=1}^K (\bar{s}_k - \bar{s})^2 - \frac{1}{M}\hat{\sigma}_{\text{intra}}^2$
16: $\hat{\sigma}_{\text{total}} \leftarrow \sqrt{(1+\frac{1}{K})\hat{\sigma}_{\text{inter}}^2 + (1+\frac{1}{KM})\hat{\sigma}_{\text{intra}}^2}$
17: $t \leftarrow \frac{s_0 - \bar{s}}{\hat{\sigma}_{\text{total}}}$
18: $u_{\text{inter}} \leftarrow (1+1/K)\hat{\sigma}_{\text{inter}}^2, \quad u_{\text{intra}} \leftarrow (1+1/KM)\hat{\sigma}_{\text{intra}}^2$
19: $v \leftarrow \dfrac{(u_{\text{inter}} + u_{\text{intra}})^2}{\frac{u_{\text{inter}}^2}{K-1} + \frac{u_{\text{intra}}^2}{K(M-1)}}$
20: Compute p-value $p \leftarrow 1 - F_t(t; v)$, where $F_t$ is the CDF of the $t$-distribution.
21: **Return** $m = 1$ if $p < \alpha$, and $m = 0$ otherwise.

---

The detailed derivation is provided in Appendix A.1. From the Proposition 3.1, the Bayesian samples mainly reduce the variance from $\sigma_{\text{intra}}^2$. Following this proposition, we can derive the rejection region for $s_0$:

$$\mathcal{R} = \{s_0 \mid s_0 > \bar{s} + t_{1-\alpha,v} \cdot \hat{\sigma}_{\text{total}}\}, \quad (13)$$

where $\hat{\sigma}_{\text{total}} = \sqrt{(1+\frac{1}{K})\hat{\sigma}_{\text{inter}}^2 + (1+\frac{1}{KM})\hat{\sigma}_{\text{intra}}^2}$ is the estimated standard deviation of the test statistic.

We further analyze the impact of the number of posterior samples $M$ on the detection performance. As seen in the variance term $(1 + \frac{1}{KM})\sigma_{\text{intra}}^2$, the contribution of the intra-model uncertainty is scaled by a factor inversely proportional to $M$. Consequently, a larger $M$ reduces the total standard deviation $\hat{\sigma}_{\text{total}}$, resulting in a tighter confidence interval for the null distribution. A smaller $\hat{\sigma}_{\text{total}}$ lowers the decision threshold $\bar{s} + t_{1-\alpha,v} \cdot \hat{\sigma}_{\text{total}}$, thus expanding the rejection region $\mathcal{R}$. Recall that the TPR for a target sample $z$ corresponds to the statistical power of the test, defined as $\text{TPR}(z) = \text{Pr}(s_0 \in \mathcal{R} \mid H_1)$, an expanded rejection region $\mathcal{R}$ directly leads to a higher power (TPR). We summarize this analysis as the following theorem:

*Table 1.* MIA performance comparison across various attacks on five datasets. TPR (%) values at 0.1% and 1% FPR are presented as percentages. Time (min) values indicate GPU minutes required for training reference models. $n$ represents the number of reference models used. **Bold** numbers are the optimal results, and underline numbers are suboptimal results. ↑ implies that larger values are better, and ↓ indicates that smaller values are better.

| Attack Method | CIFAR-10 | | | CIFAR-100 | | | Texas-100 | | | Purchase-100 | | | ImageNet | | |
|---|---|---|---|---|---|---|---|---|---|---|---|---|---|---|---|
| | TPR@FPR ↑ | | Time ↓ | TPR@FPR ↑ | | Time ↓ | TPR@FPR ↑ | | Time ↓ | TPR@FPR ↑ | | Time ↓ | TPR@FPR ↑ | | Time ↓ |
| | 0.1% | 1% | | 0.1% | 1% | | 0.1% | 1% | | 0.1% | 1% | | 0.1% | 1% | |
| Attack-P | 0.05 | 1.17 | - | 0.11 | 2.88 | - | 0.18 | 1.40 | - | 0.10 | 1.10 | - | 0.12 | 1.20 | - |
| Attack-R (n=2) | 0.24 | 3.58 | 50.77 | 0.81 | 9.06 | 52.88 | 0.12 | 1.02 | 2.06 | 0.12 | 1.24 | 4.31 | 0.31 | 5.44 | 1170.41 |
| Attack-R (n=4) | 0.26 | 3.83 | 101.54 | 1.68 | 17.37 | 105.76 | 0.12 | 2.35 | 4.12 | 0.30 | 2.03 | 8.62 | 1.25 | 11.80 | 2340.81 |
| Attack-R (n=8) | 0.48 | 5.49 | 203.09 | 3.63 | 27.35 | 211.52 | 0.12 | 5.35 | 8.23 | 0.37 | 3.39 | 17.24 | 1.83 | 13.44 | 4681.62 |
| LiRA (n=2) | 0.27 | 2.34 | 50.77 | 0.33 | 3.67 | 52.88 | 0.30 | 2.96 | 2.06 | 0.23 | 1.87 | 4.31 | 0.30 | 2.96 | 1170.41 |
| LiRA (n=4) | 0.83 | 5.42 | 101.54 | 2.89 | 19.67 | 105.76 | 0.83 | 5.42 | 4.12 | 0.37 | 3.12 | 8.62 | 0.82 | 6.91 | 2340.81 |
| LiRA (n=8) | 1.73 | 8.32 | 203.09 | 6.23 | 23.20 | 211.52 | 2.21 | 8.63 | 8.23 | 0.64 | 4.60 | 17.24 | 2.02 | 11.90 | 4681.62 |
| RMIA (n=2) | 0.99 | 4.11 | 50.77 | 3.46 | 12.86 | 52.88 | 2.30 | 8.26 | 2.06 | 0.38 | 2.33 | 4.31 | 3.30 | 10.40 | 1170.41 |
| RMIA (n=4) | 1.08 | 4.58 | 101.54 | 5.20 | 16.03 | 105.76 | 2.85 | 9.08 | 4.12 | 0.32 | 2.57 | 8.62 | 4.30 | 12.25 | 2340.81 |
| RMIA (n=8) | 1.35 | 5.55 | 203.09 | 4.51 | 16.91 | 211.52 | 2.84 | 10.12 | 8.23 | 0.39 | 2.99 | 17.24 | 4.39 | 13.54 | 4681.62 |
| **Ours (n=1)** | **2.84** | **9.48** | 25.39 | **13.11** | **35.75** | 26.44 | **3.22** | **11.81** | 1.03 | **0.71** | **4.61** | 2.16 | **4.39** | **13.59** | 585.20 |

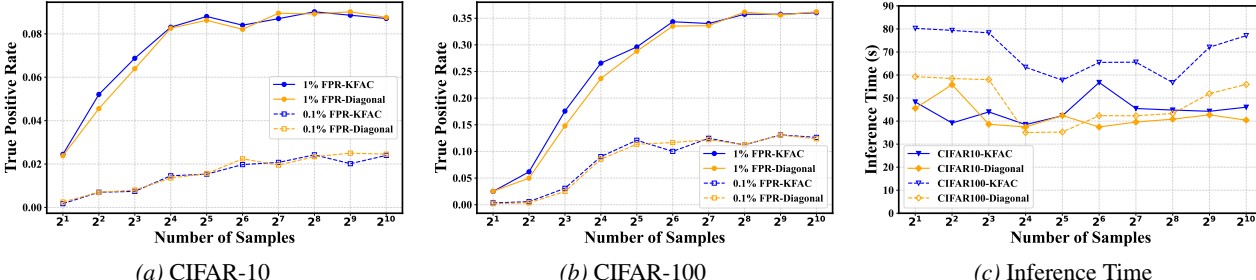

*(a)* CIFAR-10        *(b)* CIFAR-100        *(c)* Inference Time

*Figure 1.* Evaluation of BMIA under varying sample sizes and Hessian factorization methods. Panels (a) and (b) report TPR trends at 0.1% and 1% FPR, while (c) shows the corresponding inference time on CIFAR-10/100.

**Theorem 3.2.** *Let $\beta(M)$ denote the statistical power (TPR) of the membership inference with $M$ posterior samples per reference model. For the significance level $\alpha \in (0,1)$ and any $M' > M \geq 1$, we have:*

$$\beta(M') > \beta(M). \tag{14}$$

The corresponding proof is deferred to the Appendix A.2. From Theorem 3.2, a method like LiRA is equivalent to setting $M = 1$. At a limited computation budget, not enough $K$ may lead to a larger variance $\sigma_{\text{total}}^2$, thereby performing worse than our method even with one reference model.

From Proposition 3.1, we observe that the total variance consists of the intra-model variance ($\sigma_{\text{intra}}^2$) and the inter-model variance ($\sigma_{\text{inter}}^2$). Our proposed BMIA (with $K = 1$) effectively mitigates $\sigma_{\text{intra}}^2$ via Bayesian sampling. However, it is worth noting that our method naturally extends to the multi-reference setting to further reduce $\sigma_{\text{inter}}^2$. In particular, when computational resources allow training multiple reference models, we leverage a mixture Laplace approximation to develop a more powerful extension, termed the

Multi-Reference Bayesian Membership Inference Attack (**MR-BMIA**). The detailed procedure for this enhanced approach is summarized in Algorithm 2, with the variance estimator derived in Appendix C.

## 4. Experiments

### 4.1. Setups

**Datasets.** Our evaluation employs datasets across three modalities: tabular, image, and text. For tabular data, we utilize Texas100 and Purchase100 (Kaggle, 2014). Image analysis is performed on CIFAR-10/100 (Krizhevsky et al., 2009) and ImageNet-1k (Russakovsky et al., 2015). The text datasets include Emotion (Saravia et al., 2018), TREC-6 (Li & Roth, 2002), 20 Newsgroups (Lang, 1995), DBpedia (Bizer et al., 2009), and AG News (Zhang et al., 2015). Following prior work (Chen et al., 2022; Liu et al., 2024), we split each dataset into four disjoint subsets: 20% for target-model training, 20% for target-model testing, 40% as the data population for training reference models, and 20%

*Table 2.* TPR (%) and time (min) comparison on CIFAR-10/100. RMIA and our method both use a single reference model.

| Attack Method | CIFAR-10 | | | CIFAR-100 | | |
| | TPR@FPR ↑ | | Time ↓ | TPR@FPR ↑ | | Time ↓ |
| | 0.1% | 1% | | 0.1% | 1% | |
| QMIA | 0.10 | 1.23 | 306.59 | 2.58 | 15.26 | 234.92 |
| RMIA | 0.68 | 3.64 | 25.39 | 2.49 | 10.08 | 26.44 |
| **Ours** | **2.84** | **9.48** | **25.39** | **13.11** | **35.75** | **26.44** |

*Table 3.* TPR (%) comparison on CIFAR-10/100 using 64 reference models. **Ours** corresponds to the proposed MR-BMIA method summarized in Algorithm 2.

| Attack Method | CIFAR-10 | | CIFAR-100 | |
| | TPR@FPR ↑ | | TPR@FPR ↑ | |
| | 0.1% | 1% | 0.1% | 1% |
| Attack-R | 0.82 | 8.23 | 14.13 | 42.02 |
| LiRA | 4.08 | 11.74 | 13.19 | 43.33 |
| RMIA | 2.97 | 11.15 | 14.50 | 36.06 |
| **Ours** | **4.28** | **12.48** | **15.31** | **45.57** |

as the validation set for QMIA (Bertran et al., 2024). Each reference model is trained on a randomly sampled half of the reference data population. The attacker has no access to target-model training data.

**Training details.** For image and tabular datasets, we train target and reference models using SGD with weight decay 0.0005, momentum 0.9, and an initial learning rate of 0.1. We use ResNet-50 for CIFAR-10, DenseNet-121 for CIFAR-100, ResNet-50 for ImageNet, and a 4-layer MLP for Purchase-100 and Texas-100, following prior work (Nasr et al., 2019; Jia et al., 2019; Chen et al., 2022). For text datasets, we fine-tune BERT on Emotion, AG News, and DBpedia, fine-tune DistilBERT on TREC-6, and train an MLP over TF-IDF features for 20 Newsgroups. Model accuracies are reported in Appendix Table 8.

**Attack baselines** . In our experiments, we consider the following attacks: (1) Attack-P (Ye et al., 2022), which serves as the baseline attack that does not require any reference model and corresponds to the marginal attack defined in Equation (4). (2) Attack-R (Ye et al., 2022), which trains multiple reference models and then determines the instance-level threshold (defined in Equation (6)) by empirical quantile estimation. (3) LiRA (Carlini et al., 2022), which also trains multiple reference models, but computes the threshold $\tau(z)$ by Gaussian quantile approximation in the offline setting. (4) QMIA (Bertran et al., 2024), which trains a quantile regression model to directly predict $\tau(z)$. (5) RMIA (Zarifzadeh et al., 2024), which constructs an MIA score by pair-wise likelihood ratio. In this paper, we focus on the more practical offline setting.

**Implementation details.** In this paper, we employ the last-layer Laplace approximation (Kristiadi et al., 2020) combined with Kronecker-factored Approximate Curvature (KFAC) (Martens & Grosse, 2015) for Hessian estimation. The precision of the Gaussian prior over the last-layer weights is tuned via marginal likelihood maximization (MacKay, 1992). Details are provided in Appendix G.

## 4.2. Main Results

**BMIA achieves SOTA performance with a single reference model.** We first compare our method against efficient baselines, including RMIA with a single reference model and QMIA based on a quantile regression model. As shown in Table 2, BMIA achieves higher TPR at 0.1% and 1% FPRs than these baselines.

We further evaluate BMIA against a broader set of attack methods. In Table 1 and Table 6, we compare the attack performance on five datasets. Following (Bertran et al., 2024) for fair comparison, we trained 2, 4, and 8 reference models for LiRA, Attack-R, and RMIA. Though BMIA only trained one reference model, it can still achieve competitive attack results. For example, on the CIFAR-100 dataset, our method has a 54% higher true positive rate than LiRA (n=8) at a 1% false positive rate. Overall, our proposed method outperforms prior works, even with only one reference model. The log-scale ROC curves are provided in Appendix F.3.

In addition to attack effectiveness, we also measure the training time required for MIAs. For Attack-R, LiRA, RMIA, and our method, this corresponds to the time spent training reference models, while for QMIA, it refers to the time needed to train quantile regression models [1]. The inference stage in our approach, which involves fitting the LA before performing MIA, incurs minimal additional computational overhead compared to training (see Figure 1c). In Table 1, we present the training time required for each attack on five datasets, demonstrating that our method significantly reduces training time while maintaining state-of-the-art performance. Overall, we demonstrate that our method indeed achieves the purpose of efficiency.

**MR-BMIA can outperform multi-reference baselines.** In scenarios where computational resources are abundant, an attacker may train multiple reference models. We eval-

---
[1] QMIA incurs higher training costs due to expensive hyperparameter tuning. We provide a detailed analysis of its limitations relative to our method in Appendix D.

*Table 4.* Baseline comparison under architecture mismatch on CIFAR-10. The target model is ResNet-50, and the reference model is ResNet-18. We report TPR (%) at different FPRs.

| Attack Method | TPR@FPR ↑ | | | |
|---|---|---|---|---|
| | 0.1% | 0.5% | 1% | 5% |
| Attack-R | 0.55 | 3.03 | 5.30 | 18.29 |
| LiRA | 1.55 | 5.01 | 8.16 | 20.31 |
| RMIA | 1.13 | 2.98 | 5.52 | 16.81 |
| **BMIA** | **2.49** | **5.22** | **8.72** | **21.55** |

*Table 5.* TPR (%) comparison under different FPR thresholds. **Ours** corresponds to the Laplace approximation-based method.

| Method | TPR@FPR ↑ | | | |
|---|---|---|---|---|
| | 0.1% | 0.5% | 1% | 5% |
| BatchEnsemble | 0.18 | 0.61 | 1.41 | 8.40 |
| Packed Ensemble | 0.13 | 0.68 | 1.45 | 8.93 |
| Masksembles | 0.71 | 2.77 | 4.80 | 17.28 |
| SWAG | 1.26 | 4.07 | 6.61 | 19.81 |
| MC Dropout | 1.48 | 5.06 | 7.94 | 20.53 |
| **Ours** | **2.84** | **6.65** | **9.48** | **21.65** |

uate the multi-reference variant, MR-BMIA, whose procedure is summarized in Algorithm 2 and further detailed in Appendix C. Using 64 reference models trained on CIFAR-10/100, MR-BMIA achieves higher TPR than other multi-reference attacks, including Attack-R, LiRA, and RMIA, as shown in Table 3. This corroborates our analysis in Section 3.1 and shows that BMIA can derive further gains from additional reference models.

### 4.3. Ablation Studies

**Impact of sample size on attack performance.** We obtain the conditional score distribution for a given instance by sampling logits from the approximate posterior. In Figures 1a and 1b, we analyze the impact of sample size on attack performance on CIFAR-10/100 using two Hessian factorization methods: KFAC and Diagonal. The results demonstrate that increasing the sample size improves TPR, consistent with our theoretical analysis in Section 3.1. Figure 1c reports the inference time of our method, including the cost of Hessian construction, weight sampling, and p-value computation. Notably, inference time is insensitive to the sample size, as shown in Figure 1c, because all sampling operations are efficiently parallelized matrix computations.

**Sensitivity to Hessian factorization.** In general, directly computing the Hessian matrix $\Sigma$ in Equation (8) is infeasible due to its quadratic scaling with the number of parameters. Therefore, factorization methods are commonly adopted to efficiently estimate the Hessian. In particular, we consider two simple methods: Diagonal (LeCun et al., 1989; Denker & LeCun, 1990) and KFAC factorizations (Heskes, 2000; Martens & Grosse, 2015). As shown in Figures 1a and 1b, the two factorization methods exhibit no significant difference in attack results, indicating that even with the lightweight Hessian approximation method, BMIA can achieve powerful attack results. More details about the four components in BMIA are presented in Appendix G.

**Effect of Bayesian inference methods.** In this paper, we leverage the Laplace approximation to estimate the conditional distribution. To assess the impact of the underlying Bayesian inference method, we replace the Laplace approx-

imation in Algorithm 1 with several widely used alternatives, including BatchEnsemble (Wen et al., 2020), Packed-Ensembles (Laurent et al., 2023), Masksembles (Durasov et al., 2021), SWAG (Maddox et al., 2019), and Monte Carlo Dropout (Gal & Ghahramani, 2016). We evaluate the attack performance on CIFAR-10, and the results in Table 5 show that the Laplace approximation achieves the superior performance among all compared methods.

### 4.4. Robustness Analysis

**Is BMIA resilient with reference models trained on different datasets?** We investigate the resilience of BMIA when the reference model is trained on data from a different distribution than the target model's training data. This scenario is crucial in practice where attackers may lack access to sufficient non-member data from the identical distribution. Thus, we study the case where reference models are trained on a different dataset from the target model. Following (Zarifzadeh et al., 2024), we conduct attacks on CIFAR-10 with the reference model trained on CINIC-10. Figure 2a presents the TPR comparison at 0.1% and 1% FPRs for reference-model-based attacks. The results demonstrate that our attack maintains superior robustness in this setting, even with a reduced training budget.

**Is BMIA robust against OOD samples?** We evaluate the robustness of BMIA against out-of-distribution (OOD) samples. BNNs naturally model uncertainty, allowing them to flag out-of-distribution samples by exhibiting high uncertainty in their predictions (Kristiadi et al., 2020). Following the setup of previous work (Zarifzadeh et al., 2024), we evaluate our attack using a target model trained on CIFAR-10 and consider samples from CINIC-10 as OOD non-member queries. Figure 2b illustrates that BMIA surpasses the performance of alternative attacks when encountering OOD samples, even with 1/8 computation cost.

**How does BMIA perform when target and reference model architecture are mismatched?** In real-world scenarios, attackers may not have precise knowledge of the target model's architecture. We therefore evaluate our

*Table 6.* MIA performance comparison across various attacks on five text datasets. TPR (%) values at 0.1% and 1% FPR are presented as percentages. ↑ implies that larger values are better. **Bold** numbers are the optimal results, and underline numbers are suboptimal results.

| Attack Method | Emotion | | 20 Newsgroups | | AG News | | DBpedia | | TREC-6 | |
|---|---|---|---|---|---|---|---|---|---|---|
| | TPR@FPR ↑ | | TPR@FPR ↑ | | TPR@FPR ↑ | | TPR@FPR ↑ | | TPR@FPR ↑ | |
| | 0.1% | 1% | 0.1% | 1% | 0.1% | 1% | 0.1% | 1% | 0.1% | 1% |
| Attack-P | 0.18 | 1.40 | 0.19 | 1.43 | 0.13 | 1.17 | 0.10 | 1.00 | 0.08 | 0.25 |
| Attack-R (n=2) | 0.10 | 1.00 | 0.24 | 2.43 | 0.13 | 1.33 | 0.10 | 1.04 | 0.15 | 1.47 |
| Attack-R (n=4) | 0.10 | 1.00 | 0.31 | 3.10 | 0.15 | 1.48 | 0.10 | 1.05 | 0.16 | 1.59 |
| Attack-R (n=8) | 0.27 | 2.69 | 0.40 | 4.00 | 0.17 | 1.73 | 0.11 | 1.05 | 0.17 | 1.67 |
| LiRA (n=2) | 0.10 | 0.96 | 0.50 | 5.01 | 0.18 | 1.84 | 0.11 | 1.06 | 0.20 | 1.99 |
| LiRA (n=4) | 0.13 | 1.33 | 1.76 | 17.63 | 0.69 | 4.09 | 0.11 | 1.12 | 0.58 | 5.21 |
| LiRA (n=8) | 1.55 | 4.73 | 5.48 | 30.32 | 1.93 | 4.89 | 0.26 | 1.24 | 0.77 | 5.13 |
| RMIA (n=2) | 0.20 | 1.21 | 2.73 | 8.32 | 1.38 | 3.76 | 0.42 | 1.50 | 0.67 | 4.69 |
| RMIA (n=4) | 0.18 | 1.25 | 3.24 | 11.91 | 1.48 | 3.84 | 0.43 | 1.54 | 0.68 | 5.09 |
| RMIA (n=8) | 0.95 | 4.73 | 2.83 | 11.57 | 1.61 | 4.10 | 0.47 | 1.55 | 1.88 | 5.44 |
| **Ours (n=1)** | **1.75** | **6.28** | **7.90** | **33.47** | **2.21** | **5.71** | **0.61** | **1.89** | **2.69** | **6.89** |

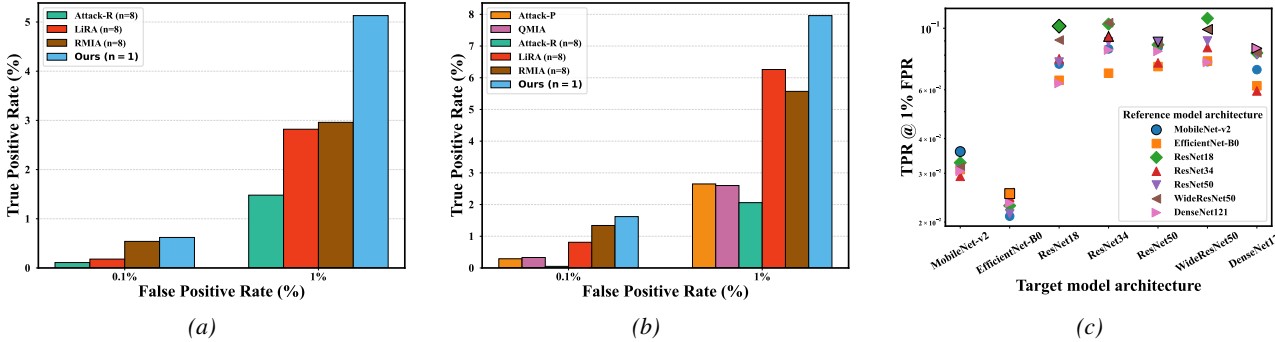

*Figure 2.* Performance of BMIA across three scenarios: (a) Reference models are trained on the CINIC-10 dataset. (b) Non-member queries are composed of OOD samples sourced from the CINIC-10 dataset. (c) Reference and target model architectures are mismatched.

attacks under mismatched model structures. Figure 2c presents the TPR at a 1% false positive rate, comparing target models and reference models trained with various architectures on CIFAR-10. The architectures include ResNet-18, ResNet-34, ResNet-50, Wide-ResNet-50, DenseNet-121, MobileNet-V2 (Howard, 2017), and EfficientNet-B0 (Koonce & Koonce, 2021), all trained with consistent optimization settings. In general, our attack performs best when the reference and target models share the same architecture. However, in certain scenarios, mismatched but structurally similar architectures can also yield the best results. For instance, when the target model is ResNet-34, the best performance is achieved with a reference model trained on ResNet-18, highlighting the robustness of our method. Table 4 further provides a baseline comparison in a representative mismatch setting where the target model is ResNet-50 and the reference model is ResNet-18.

## 5. Related Work

### 5.1. Membership Inference Attacks.

MIAs are commonly employed as auditing tools to assess the privacy risk of machine learning models (Homer et al., 2008; Jayaraman & Evans, 2019; Jagielski et al., 2020a; Nasr et al., 2021). Additionally, MIAs are utilized in extraction attacks on generative models (Carlini et al., 2021) and are integral to evaluating machine unlearning (Zhang et al., 2023). Beyond supervised classification, MIAs have been extended to additional fields (Li, 2024), such as graph embedding models (Duddu et al., 2020; Wang & Wang, 2023), graph neural network (Olatunji et al., 2021; Liu et al., 2022; Conti et al., 2022; Lassila et al., 2025), contrastive learning (Liu et al., 2021; Ko et al., 2023; Wang & Wang, 2024), language models (Mireshghallah et al., 2022; Mattern et al., 2023; Wang et al., 2024; Wen et al., 2024; Li et al., 2024; Fu et al., 2024; Zhai et al., 2024; Zhang et al., 2025a;b; Fu et al., 2025; Liu et al., 2026; Yi & Li, 2026), vision language model (Li et al., 2024; LIU et al., 2025), generative models

(Chen et al., 2020; Pang et al., 2023; van Breugel et al., 2023; Hu & Pang, 2021; Li* et al., 2024; Tang et al., 2024; Dubiński et al., 2024) and recommend system (Chi et al., 2024; Zhong et al., 2024; He et al., 2025)

First introduced for machine learning models by Shokri et al. (2017), training reference models (also known as shadow models) is a popular method for mimicking the target model's behavior. This approach is widely used in various MIA scenarios, such as black-box setting (Hisamoto et al., 2020; Song & Mittal, 2021; Hu et al., 2022), white-box setting (Nasr et al., 2019; Sablayrolles et al., 2019; Leino & Fredrikson, 2020; Leemann et al., 2023), metric-based attacks (Yeom et al., 2018; Salem et al., 2019; Liu et al., 2019; Song & Mittal, 2021) and label-only attack (Choquette-Choo et al., 2021; Li & Zhang, 2021; Wu et al., 2024; Peng et al., 2024). In metric-based attacks, shadow models are typically used to choose a proper threshold (Song & Mittal, 2021; Chen et al., 2022), classifying them as marginal attacks. To enhance these attacks, many approaches train multiple reference models to account for per-example hardness (Sablayrolles et al., 2019; Carlini et al., 2022; Ye et al., 2022; Hu et al., 2022), but at the cost of substantial computational overhead. Alternatively, Bertran et al. (2024) use a quantile regression model to directly predict the threshold. However, this approach is limited to modeling aleatoric uncertainty and does not explicitly account for epistemic uncertainty, as discussed in Appendix D. In this work, our method achieves conditional attacks using just one reference model with better uncertainty quantification.

**5.2. Bayesian Neural Network.**

Bayesian Neural Network (BNN) (Gelman et al., 1995) extends standard neural networks by incorporating probabilistic inference, enabling uncertainty estimation in both the model parameters and predictions. To achieve scalable uncertainty estimation without fully Bayesian training, a number of practical BNN approximations have been proposed. One line of work relies on stochastic approximations to Bayesian inference. For example, SWAG (Maddox et al., 2019) captures parameter uncertainty by fitting a Gaussian distribution to SGD trajectories, enabling efficient posterior sampling at inference time. Similarly, Monte Carlo Dropout (MC Dropout) (Gal & Ghahramani, 2016) interprets dropout as approximate Bayesian inference and estimates predictive uncertainty through multiple stochastic forward passes. Another direction leverages ensemble-style parameter sharing to reduce computational overhead. Specifically, BatchEnsemble (Wen et al., 2020) employs rank-1 factors to scale weights, Masksembles (Durasov et al., 2021) utilizes fixed binary masks to induce diversity, and Packed-Ensembles (Laurent et al., 2023) applies group convolutions to parallelize ensemble members. While efficient, these approaches typically require specific training procedures

or structural modifications. In this paper, we leverage the Laplace Approximation, a scalable and post-hoc method that can be directly applied to a pre-trained model without modifying the training procedure or architecture.

# 6. Conclusion

In this paper, we propose Bayesian Membership Inference Attack (**BMIA**), a scalable and effective method for performing conditional attacks. Our approach begins by training a reference model through standard DNN training. We then apply the Laplace Approximation to estimate the posterior distribution of the model parameters, enabling us to compute the conditional score distribution under the hypothesis that the queried data point is a non-member. This allows BMIA to perform conditional attacks using only a single reference model while providing a more accurate estimation of the conditional distribution. Furthermore, we provide a theoretical analysis demonstrating how Bayesian sampling enhances MIAs through the lens of variance reduction. This analysis naturally extends to an enhanced variant, Multi-Reference BMIA (MR-BMIA), which further boosts performance when additional reference models are available. Extensive experiments demonstrate that BMIA achieves state-of-the-art performance compared to prior methods.

# Acknowledgements

This research is supported by Guangdong Basic and Applied Basic Research Foundation (Grant No. 2026A1515011367). This project is also supported by the Jiangsu Provincial Key Discipline Construction Project (Statistics) and the open project of the Joint Lab for Statistics and Finance (Grant No. 2025JLSF101). We gratefully acknowledge the support of the Center for Computational Science and Engineering at the Southern University of Science and Technology for our research.

# Impact Statement

This paper presents work whose goal is to advance the field of Machine Learning. There are many potential societal consequences of our work, none of which we feel must be specifically highlighted here.

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

# A. Proofs

## A.1. Proof of Proposition 3.1

First, we derive the variance of the aggregated score $\bar{s}$:

$$\text{Var}(\bar{s}) = \text{Var}\left(\frac{1}{K}\sum_{k=1}^{K}\frac{1}{M}\sum_{m=1}^{M}s_{k,m}\right)$$

$$= \mathbb{E}\left[\text{Var}\left(\bar{s}\mid\{\mathcal{D}_k\}_{k=1}^{K}\right)\right] + \text{Var}\left(\mathbb{E}\left[\bar{s}\mid\{\mathcal{D}_k\}_{k=1}^{K}\right]\right)$$

(Applying Law of Total Variance conditioned on datasets)

$$= \mathbb{E}\left[\text{Var}\left(\frac{1}{KM}\sum_{k=1}^{K}\sum_{m=1}^{M}s_{k,m}\;\Big|\;\{\mathcal{D}_k\}\right)\right]$$

$$+ \text{Var}\left(\frac{1}{K}\sum_{k=1}^{K}\mathbb{E}\left[\frac{1}{M}\sum_{m=1}^{M}s_{k,m}\;\Big|\;\mathcal{D}_k\right]\right)$$

$$= \mathbb{E}\left[\frac{1}{(KM)^2}\sum_{k=1}^{K}\sum_{m=1}^{M}\text{Var}(s_{k,m}|\mathcal{D}_k)\right]$$

$$+ \text{Var}\left(\frac{1}{K}\sum_{k=1}^{K}\mathbb{E}[s|\mathcal{D}_k]\right)$$

(Using conditional independence of samples)

$$= \frac{1}{K^2M^2}\sum_{k=1}^{K}\sum_{m=1}^{M}\mathbb{E}[\text{Var}(s|\mathcal{D}_k)]$$

$$+ \frac{1}{K^2}\sum_{k=1}^{K}\text{Var}(\mathbb{E}[s|\mathcal{D}_k])$$

$$= \frac{1}{K^2M^2}\cdot(K\cdot M)\cdot\sigma_{\text{intra}}^2 + \frac{1}{K^2}\cdot K\cdot\sigma_{\text{inter}}^2$$

$$= \frac{\sigma_{\text{intra}}^2}{K\cdot M} + \frac{\sigma_{\text{inter}}^2}{K}$$

Under $H_0$, $s_0$ follows the same distribution as the reference scores $s_{k,m}$. Furthermore, since the target sample $s_0$ is independent of the reference datasets used to compute $\bar{s}$, the covariance term is zero. Thus:

$$\text{Var}(s_0 - \bar{s}) = \text{Var}(s_0) + \text{Var}(\bar{s})$$

$$= \left(\sigma_{\text{inter}}^2 + \sigma_{\text{intra}}^2\right) + \left(\frac{\sigma_{\text{inter}}^2}{K} + \frac{\sigma_{\text{intra}}^2}{KM}\right) \tag{15}$$

$$= \left(1 + \frac{1}{K}\right)\sigma_{\text{inter}}^2 + \left(1 + \frac{1}{KM}\right)\sigma_{\text{intra}}^2.$$

## A.2. Proof of Theorem 3.2

*Proof.* Let $\tau(M) = \bar{s} + t_{1-\alpha,v}\cdot\sigma_{\text{total}}(M)$ be the decision threshold, where

$$\sigma_{\text{total}}(M) = \sqrt{(1+\frac{1}{K})\sigma_{\text{inter}}^2 + (1+\frac{1}{KM})\sigma_{\text{intra}}^2}.$$

Given $M' > M$, since the intra-model variance $\sigma_{\text{intra}}^2 > 0$, we have the inequality:

$$\frac{1}{KM'} < \frac{1}{KM} \implies \sigma_{\text{total}}(M') < \sigma_{\text{total}}(M). \tag{16}$$

Since the critical value $t_{1-\alpha,v} > 0$ for $\alpha < 0.5$, the reduction in standard deviation leads to a strictly lower threshold:

$$\tau(M') < \tau(M). \tag{17}$$

The power is defined as the tail probability $\beta(M) = \Pr(s_0 > \tau(M) \mid H_1)$. Since the cumulative distribution function is non-decreasing, a lower threshold implies a larger rejection region:

$$\Pr(s_0 > \tau(M') \mid H_1) > \Pr(s_0 > \tau(M) \mid H_1). \tag{18}$$

Thus, $\beta(M') > \beta(M)$. $\qquad\square$

### A.3. Linear-Regression Interpretation

A Gaussian linear model gives a clean interpretation of the relationship between training multiple reference models and sampling from the posterior of a single reference model. Consider

$$y = x^\top w^\star + \epsilon, \quad \epsilon \sim \mathcal{N}(0, \sigma^2),$$

and let $\Sigma_x = \mathbb{E}[xx^\top]$. For a dataset of size $N$, denote its design matrix by $X$. If we train reference models on independently resampled datasets, the resulting estimators $\hat{w}_k$ vary because the underlying training data vary. Under standard regularity conditions for linear regression, these estimators are asymptotically centered at $w^\star$ and satisfy

$$N \operatorname{Cov}(\hat{w}_k) \to \sigma^2 \Sigma_x^{-1}.$$

Now consider a single realized reference dataset $(X_1, Y_1)$. In the Gaussian linear model with a flat prior, the posterior over weights is

$$\tilde{w}_m \mid X_1, Y_1 \sim \mathcal{N}\left(\hat{w}_1, \sigma^2(X_1^\top X_1)^{-1}\right),$$

so its conditional covariance satisfies

$$N \operatorname{Cov}(\tilde{w}_m \mid X_1, Y_1) = N\sigma^2(X_1^\top X_1)^{-1} \xrightarrow{p} \sigma^2 \Sigma_x^{-1}.$$

Thus, conditional on one realized reference dataset, posterior sampling from a single reference model matches the leading-order local covariance of the estimators that would be obtained by repeated retraining.

The two procedures are still not identical because their centers differ. Repeated retraining is asymptotically centered at $w^\star$ under data resampling, whereas posterior samples from one reference model are conditionally centered at the dataset-specific estimator $\hat{w}_1$. The center gap is

$$\hat{w}_1 - w^\star = (X_1^\top X_1)^{-1} X_1^\top \epsilon_1,$$

which is a realization-dependent $O_p(N^{-1/2})$ fluctuation induced by the specific reference dataset. For linear predictions, and more generally for smooth scores by a delta-method argument, these parameter covariance statements translate into the corresponding leading-order score covariance. This distinction explains why Proposition 3.1 retains the inter-model variance term: posterior sampling captures the within-model uncertainty around a fixed reference dataset, but it does not remove the dataset-to-dataset fluctuation of the center that would arise from retraining on multiple datasets.

## B. Theoretical Analysis of Conditional Attacks

Our method trains one reference model and then uses Bayesian inference to perform the conditional attack, as proposed for considering per-example hardness (Carlini et al., 2022; Ye et al., 2022). However, previous work does not formally analyze why the conditional attack is effective. In this paper, we address this gap with a theoretical explanation.

Let $P$ denote the random variable representing the value of $s(x, y; w)$ for $(x, y) \in \mathcal{S}$. The overall privacy risk for the attack defined in Equation (4) depends on the distinction between distribution $P$ and $Q$. Specifically, assuming that $P$ and $Q$ follow a Gaussian distribution, a common assumption in hinge score distributions (Bertran et al., 2024; Carlini et al., 2022; Zarifzadeh et al., 2024), we derive how the distribution shape influences the power of MIAs. Here, $\mathrm{TPR}_m$ denotes the true positive rate for the marginal attack, as defined in Equation (4).

**Proposition B.1.** *Suppose $P$ and $Q$ follow the normal distribution such that $P \sim \mathcal{N}(\mu_{\mathcal{S}}, \sigma_{\mathcal{S}}^2)$ and $Q \sim \mathcal{N}(\mu_{\mathcal{D}}, \sigma_{\mathcal{D}}^2)$. Then the true positive rate (TPR) at the low FPR $\alpha$ is:*

$$\text{TPR}_{\text{m}} = \Pr(\mathcal{A} = 1 | m = 1) = \Phi\left(\frac{\mu_{\mathcal{S}} - \mu_{\mathcal{D}} + \Phi^{-1}(\alpha)\sigma_{\mathcal{D}}}{\sigma_{\mathcal{S}}}\right), \tag{19}$$

*where $\Phi(\cdot)$ is the cumulative distribution function of standard normal distribution.*

*Proof.* Let $S$ denote the random variables representing the scores obtained by the target model for the test query. Since $\tau$ is determined by Equation (4), we have

$$\alpha = \Pr\left(Q \geqslant \tau_{\alpha}\right) \tag{20}$$

$$= \Pr\left(\frac{Q - \mu_{\mathcal{D}}}{\sigma_{\mathcal{D}}} \geqslant \frac{\tau_{\alpha} - \mu_{\mathcal{D}}}{\sigma_{\mathcal{D}}}\right) \tag{21}$$

$$= \Phi\left(\frac{\mu_{\mathcal{D}} - \tau_{\alpha}}{\sigma_{\mathcal{D}}}\right) \tag{22}$$

Hence, $\tau_{\alpha} = \mu_{\mathcal{D}} - \Phi^{-1}(\alpha)\sigma_{\mathcal{D}}$. Then the true positive rate (TPR) for the marginal attack at the low FPR $\alpha$ is:

$$\text{TPR}_{\text{m}} = \Pr(\mathcal{A} = 1 | m = 1) \tag{23}$$

$$= \Pr(S \geqslant \tau | m = 1) \tag{24}$$

$$= \Pr\left(\frac{S - \mu_{\mathcal{S}}}{\sigma_{\mathcal{S}}} \geqslant \frac{\tau_{\alpha} - \mu_{\mathcal{S}}}{\sigma_{\mathcal{S}}} \;\middle|\; m = 1\right) \tag{25}$$

$$= 1 - \Phi\left(\frac{\tau_{\alpha} - \mu_{\mathcal{S}}}{\sigma_{\mathcal{S}}}\right) \tag{26}$$

$$= \Phi\left(\frac{\mu_{\mathcal{S}} - \tau_{\alpha}}{\sigma_{\mathcal{S}}}\right) \tag{27}$$

$$= \Phi\left(\frac{\mu_{\mathcal{S}} - \mu_{\mathcal{D}} + \Phi^{-1}(\alpha)\sigma_{\mathcal{D}}}{\sigma_{\mathcal{S}}}\right) \tag{28}$$

$\square$

From Proposition B.1, lower $\sigma_{\mathcal{D}}$ and $\sigma_{\mathcal{S}}$ result in sharper, more distinct distributions, which improve separability and increase the TPR at a fixed FPR. Notably, for $\alpha > 0.5$ where $\Phi^{-1}(\alpha) > 0$, a higher $\sigma_{\mathcal{D}}$ leads to a higher TPR, which cannot reliably serve as a metric for privacy leakage.

Based on Proposition B.1, we derive the proposition that conditional attack is more powerful than marginal attack. For simplicity, we denote the data pair $(x, y)$ as $z$. Considering the conditional score distribution $Q|z$ and $P|z$, we establish the following proposition:

**Proposition B.2.** *Suppose the conditional distribution $P|z$ and $Q|z$ follow the normal distribution such that $P|z \sim \mathcal{N}(\mu_{\mathcal{S}}(z), \sigma_{\mathcal{S}}^2(z))$ and $Q|z \sim \mathcal{N}(\mu_{\mathcal{D}}(z), \sigma_{\mathcal{D}}^2(z))$. Then we have:*

$$\text{TPR}_{\text{c}} = \mathbb{E}_{p(z)}\left[\Phi\left(\frac{\mu_{\mathcal{S}}(z) - \mu_{\mathcal{D}}(z) + \Phi^{-1}(\alpha)\sigma_{\mathcal{D}}(z)}{\sigma_{\mathcal{S}}(z)}\right)\right]$$

$$\geqslant \Phi\left(\frac{\mu_{\mathcal{S}} - \mu_{\mathcal{D}} + \Phi^{-1}(\alpha)\sigma_{\mathcal{D}}}{\sigma_{\mathcal{S}}}\right) = \text{TPR}_{\text{m}},$$

*where $\text{TPR}_{\text{c}}$ is the TPR for conditional attack.*

*Proof.* Consider a sufficiently small value of $\alpha$ such that the true positive rate (TPR) at $z$ satisfies $\text{TPR}(z) < 0.5$. Under

this condition, the function $\Phi$ is convex. Then we have:

$$\text{TPR}_c = \mathbb{E}_{p(z)}\text{TPR}(z) \tag{29}$$

$$= \mathbb{E}_{p(z)}\left[\Phi\left(\frac{\mu_{\mathcal{S}}(z) - \mu_{\mathcal{D}}(z) + \Phi^{-1}(\alpha)\sigma_{\mathcal{D}}(z)}{\sigma_{\mathcal{S}}(z)}\right)\right] \tag{30}$$

$$\geqslant \Phi\left[\mathbb{E}\left(\frac{\mu_{\mathcal{S}}(z) - \mu_{\mathcal{D}}(z) + \Phi^{-1}(\alpha)\sigma_{\mathcal{D}}(z)}{\sigma_{\mathcal{S}}(z)}\right)\right] \quad \text{(By Jensen's inequality)} \tag{31}$$

$$= \Phi\left[(\mu_{\mathcal{S}} - \mu_{\mathcal{D}} + \Phi^{-1}(\alpha)\mathbb{E}_{p(z)}\sigma_{\mathcal{D}}(z))\mathbb{E}_{p(z)}(\frac{1}{\sigma_{\mathcal{S}}(z)})\right] \tag{32}$$

$$\geqslant \Phi\left[(\mu_{\mathcal{S}} - \mu_{\mathcal{D}} + \Phi^{-1}(\alpha)\sigma_{\mathcal{D}})\mathbb{E}_{p(z)}(\frac{1}{\sigma_{\mathcal{S}}(z)})\right] \quad \text{(By variance decomposition inequality)} \tag{33}$$

$$\geqslant \Phi\left(\frac{\mu_{\mathcal{S}} - \mu_{\mathcal{D}} + \Phi^{-1}(\alpha)\sigma_{\mathcal{D}}}{\mathbb{E}_{p(z)}\sigma_{\mathcal{S}}(z)}\right) \quad \text{(By Jensen's inequality)} \tag{34}$$

$$\geqslant \Phi\left(\frac{\mu_{\mathcal{S}} - \mu_{\mathcal{D}} + \Phi^{-1}(\alpha)\sigma_{\mathcal{D}}}{\sigma_{\mathcal{S}}}\right) \quad \text{(By variance decomposition inequality)} \tag{35}$$

$$= \text{TPR}_m. \tag{36}$$

$\square$

Proposition B.2 holds due to inequality $\mathbb{E}[\text{Var}(Q|z)] < \text{Var}(Q)$, which means the conditional distributions are generally more distinct.

By BMIA, we efficiently estimate the conditional distribution $Q|z$. In Figure 3, we show a non-member sample with a hinge score above 95% of non-members to demonstrate why conditional attacks succeed. Attack-P determines the threshold by marginal distribution, which fails to reject the easy non-member example with higher score. QMIA (Bertran et al., 2024) employs quantile regression for $Q|z$, but ignores epistemic uncertainty, leading to underestimated variance, especially for samples in low-density regions. This prevents QMIA from rejecting samples with high hinge scores but also high variance, which require a higher threshold. LiRA (Carlini et al., 2022) trains multiple reference models and computes the mean and variance of the hinge score for a given sample, providing an oracle estimate of the conditional distribution. However, this approach incurs a significant training cost. Our method BMIA leverages the LA to estimate the conditional distribution by just one reference model, which achieves efficiency and accuracy simultaneously.

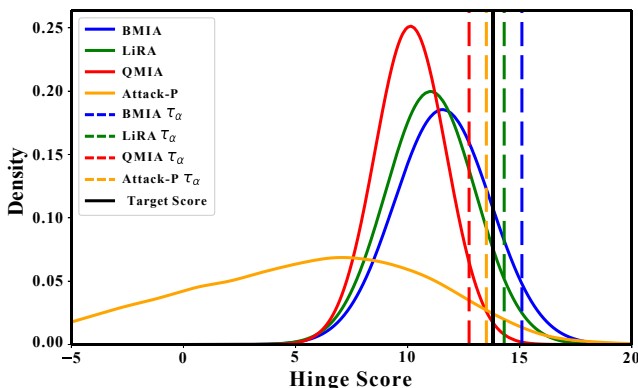

*Figure 3.* Conditional probability distribution function for a non-member sample with a higher hinge score over the non-member world, as estimated by our attack BMIA, LiRA (Carlini et al., 2022), QMIA (Bertran et al., 2024), and the marginal attack (Ye et al., 2022). Dashed lines are the estimated threshold $\tau_\alpha$ ($\alpha = 5\%$) defined in Equations (4) and (6). The target score refers to the score obtained from the target model.

## C. Scaling BMIA with Multiple Reference Models

In Algorithm 1, we prioritized computational efficiency by employing a single reference model ($K = 1$). This approach effectively captures the *intra-model variance* ($\sigma_{\text{intra}}^2$) through Bayesian inference. However, as established in Proposition 3.1, the total variance of the score distribution also includes the *inter-model variance* ($\sigma_{\text{inter}}^2$), which arises from the stochasticity of the reference dataset sampling.

To further scale the attack performance and minimize $\sigma_{\text{inter}}^2$, we extend BMIA to a multi-reference setting, named Multi-Reference Bayesian Membership Inference Attack (**MR-BMIA**). Specifically, we leverage the mixture Laplace approxima-

tion (Eschenhagen et al., 2021):

$$p_{\text{MoLA}}(w) \approx \frac{1}{K} \sum_{i=1}^{K} \mathcal{N}(w; \hat{w}_i, \Sigma_i), \tag{37}$$

where each component $\hat{w}_i$ represents a local Laplace approximation derived from a reference model trained on an independent dataset subset $\mathcal{D}_j$.

Following the empirical observation that standardized scores of non-members are approximately normally distributed (Carlini et al., 2021), we formulate the membership inference as a one-sided hypothesis test. Under the null hypothesis, the difference between the target score $s_0$ and the ensemble mean $\bar{s}$ follows a Gaussian distribution. Thus, the test statistic follows a Student's $t$-distribution defined as:

$$t = \frac{s_0 - \bar{s}}{\hat{\sigma}_{\text{total}}}, \quad \text{with} \quad \hat{\sigma}_{\text{total}} = \sqrt{\left(1 + \frac{1}{K}\right)\hat{\sigma}_{\text{inter}}^2 + \left(1 + \frac{1}{KM}\right)\hat{\sigma}_{\text{intra}}^2}. \tag{38}$$

Specifically, the intra-model variance $\sigma_{\text{intra}}^2$ and inter-model variance $\sigma_{\text{inter}}^2$ are estimated by:

$$\hat{\sigma}_{\text{intra}}^2 = \frac{1}{K(M-1)} \sum_{k=1}^{K} \sum_{m=1}^{M} (s_{k,m} - \bar{s}_k)^2, \tag{39}$$

$$\hat{\sigma}_{\text{inter}}^2 = \frac{1}{K-1} \sum_{k=1}^{K} (\bar{s}_k - \bar{s})^2 - \frac{1}{M} \hat{\sigma}_{\text{intra}}^2, \tag{40}$$

where $\bar{s}_k = \frac{1}{M} \sum_{m=1}^{M} s_{k,m}$ denotes the average score of the $k$-th reference model.

A critical component of this test is the degrees of freedom $v$, which determines the tail thickness of the null distribution. Specifically, we compute $v$ using the Welch-Satterthwaite approximation (Satterthwaite, 1946). Let $u_{\text{inter}} = (1 + 1/K)\hat{\sigma}_{\text{inter}}^2$ and $u_{\text{intra}} = (1 + 1/KM)\hat{\sigma}_{\text{intra}}^2$ denote the weighted variance contributions. The degrees of freedom is calculated as:

$$v \approx \frac{(u_{\text{inter}} + u_{\text{intra}})^2}{\frac{u_{\text{inter}}^2}{K-1} + \frac{u_{\text{intra}}^2}{K(M-1)}}. \tag{41}$$

Given this, we could conduct the t-test:

$$\mathcal{R} = \left\{ s_0 \mid s_0 > \bar{s} + t_{1-\alpha,v} \cdot \hat{\sigma}_{\text{total}} \right\},$$

Table 7 reports how attack performance changes as the number of reference models increases. Both LiRA and MR-BMIA benefit from larger $K$, but the gains diminish as $K$ grows. This trend is consistent with the variance decomposition in Proposition 3.1: increasing $K$ reduces the inter-model variance term, while the remaining intra-model variance and finite-sample effects limit the marginal improvement from adding more reference models.

*Table 7.* Scaling attack performance with the number of reference models $K$. We report TPR (%) at 1% FPR.

| Attack Method | $K = 8$ | $K = 16$ | $K = 32$ | $K = 64$ | $K = 128$ |
|---|---|---|---|---|---|
| LiRA | 8.32 | 10.15 | 11.28 | 11.71 | 11.92 |
| **MR-BMIA** | **10.14** | **11.40** | **11.87** | **12.35** | **12.77** |

# D. Bayesian Sampling Captures Epistemic Uncertainty Beyond Quantile Regression

In this paper, we leverage Bayesian sampling to estimate the conditional distribution. An alternative approach is to train a quantile regression model that directly estimates the conditional threshold (Bertran et al., 2024), enabling the conditional attack through a single model.

However, the quantile regression is designed to capture the aleatoric uncertainty (Tagasovska & Lopez-Paz, 2019) but fails to quantify the epistemic uncertainty. Aleatoric uncertainty arises from inherent randomness or noise in the data or

*Table 8.* Accuracy results across different datasets

| Dataset | CIFAR-10 | CIFAR-100 | Purchase-100 | Texas-100 | ImageNet | Emotion | 20 Newsgroups | AG News | DBpedia | TREC-6 |
|---|---|---|---|---|---|---|---|---|---|---|
| Train Accuracy | 99.83 | 99.98 | 100.00 | 99.93 | 95.74 | 98.86 | 97.64 | 99.86 | 100.00 | 96.13 |
| Test Accuracy | 80.26 | 44.48 | 89.25 | 52.76 | 63.78 | 90.12 | 67.27 | 93.08 | 99.16 | 90.08 |

the system being modeled. It is also called irreducible uncertainty because it cannot be reduced with more data or better models. Epistemic uncertainty arises from insufficient data or inadequate model complexity, which is also called knowledge uncertainty (Der Kiureghian & Ditlevsen, 2009).

Consider a regression model over a continuous-valued target $y \in \mathbb{R}$ such that $y = f(x) + \epsilon$, where $\epsilon$ is the noise. The total uncertainty can be decomposed into epistemic uncertainty and aleatoric uncertainty using the law of total variance (Depeweg et al., 2018; Malinin et al., 2021):

$$\underbrace{\mathrm{Var}(y \mid x)}_{\text{Total Uncertainty}} = \underbrace{\mathrm{Var}_{p(w|\mathcal{D})}\left[\mathbb{E}(y \mid x, w)\right]}_{\text{Epistemic Uncertainty}} + \underbrace{\mathbb{E}_{p(w|\mathcal{D})}\left[\mathrm{Var}(y \mid x, w)\right]}_{\text{Aleatoric Uncertainty}}. \tag{42}$$

Note that in Equation (42), $\mathrm{Var}_{p(w|\mathcal{D})}\left[\mathbb{E}(y \mid x, w)\right]$ represents the variance of the expected value $\mathbb{E}(y \mid x, w)$ when $w$ is drawn from the posterior distribution $p(w \mid \mathcal{D})$. This term captures the epistemic uncertainty, which is the uncertainty arising from the model parameters $w$ and their distribution. It does not account for the noise $\epsilon$ since it ignores the uncertainty from the noise $\epsilon$. In contrast, the term $\mathbb{E}_{p(w|\mathcal{D})}\left[\mathrm{Var}(y \mid x, w)\right]$ represents the expected value of the variance of $y$ conditioned on $x$ and $w$, averaged over the distribution $p(w \mid \mathcal{D})$. This term quantifies the aleatoric uncertainty, which arises from the inherent noise in the data.

**The quantile regression fails to capture epistemic uncertainty.** From Equation (42), epistemic uncertainty is captured by the variability in model parameters, which arises from having limited data or insufficient prior knowledge. However, quantile regression (QR) provides a point estimate for each quantile but does not account for the uncertainty in those estimates. Even if the data are sparse, QR will still give a specific quantile estimate without adjusting for how uncertain that estimate is, given the underlying model or parameter uncertainties.

Consider a simple regression problem, where $y = \sin(1.2x) + \epsilon$ with $\epsilon \sim \mathcal{N}(0, 0.01)$, the training dataset for $x$ is drawn from a Gaussian mixture distributions with mean parameters $\{\mu_1 = -3, \mu_2 = 3\}$, variance parameters $\{\sigma_1 = 0.8, \sigma_2 = 0.8\}$ and the mixing coefficients are all $\frac{1}{2}$. As shown in Figure 4, many points are concentrated on the edges, with few in the center, reflecting the lower

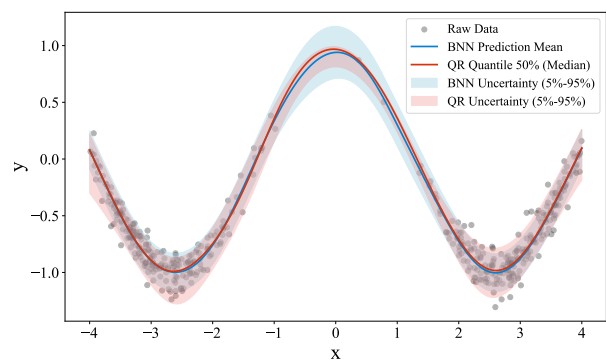

*Figure 4.* Results on a toy regression task with non-uniform distribution data (gray points). The shaded regions represent 90% prediction intervals estimated by BNN and quantile regression.

density in the trough region. We train a quantile regression model and a BNN to estimate the median (solid line) and 90% prediction interval. Quantile regression fails to capture epistemic uncertainty, particularly in the low-density center, where its intervals remain narrow despite limited data. In contrast, BNN produces wider uncertainty bounds in such regions, better reflecting the total uncertainty.

## E. Training Details

### E.1. Required Resources

All models and methods were implemented using the PyTorch framework. Our experiments were performed using NVIDIA RTX 4090 and NVIDIA L40 GPUs. The associated training and inference costs are presented in Table 1 and Figure 1c, respectively.

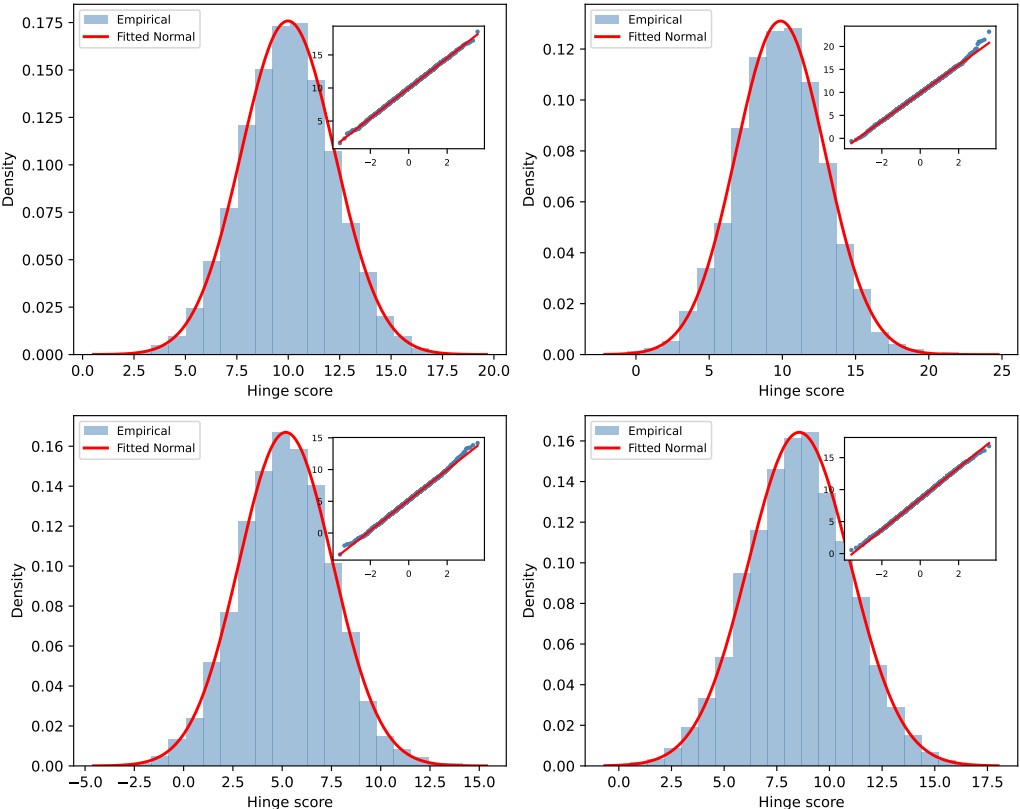

*Figure 5.* Distribution diagnostics of hinge scores for randomly selected samples. Each panel shows a histogram with a fitted normal curve and an embedded QQ-plot.

## F. Supplementary Empirical Results

### F.1. Empirical Diagnostics for the Gaussian Assumption

Our one-sample $t$-test relies on the standard approximation that the non-member score distribution is close to Gaussian. To examine this assumption, we randomly select query points and compute their hinge scores across reference models. As shown in Figure 5, the empirical hinge-score distributions are well aligned with fitted normal curves, and the embedded QQ-plots show no severe departure from normality in the evaluated setting. We further plot the empirical distribution of the resulting $t$-statistics in Figure 6, which supports the use of the Student's $t$ approximation for the test statistic.

When the score distribution is strongly non-Gaussian or heavy-tailed, the constructed statistic may no longer follow a Student's $t$-distribution exactly, and the nominal $p$-values may be miscalibrated. In such cases, one can treat the statistic as an MIA score and calibrate it with a held-out non-member calibration set, for example by computing conformal $p$-values. This calibration preserves the attack score while providing finite-sample control of the false positive rate under the calibration distribution.

### F.2. Relation to Hypothesis-Testing Privacy Views

BMIA is most closely related to hypothesis-testing views of privacy, such as $f$-differential privacy and Gaussian differential privacy (Dong et al., 2022), because these frameworks also characterize privacy through tradeoffs between false positives and true positives. This connection is conceptual rather than definitional. $f$-DP and GDP define mechanism-level privacy guarantees through the full tradeoff function between neighboring datasets, whereas BMIA constructs a concrete membership inference test against a fixed trained model and reports empirical attack power at low false positive rates. Thus, BMIA can be viewed as an attack-side instantiation of a hypothesis-testing perspective, not as a privacy definition for a randomized mechanism.

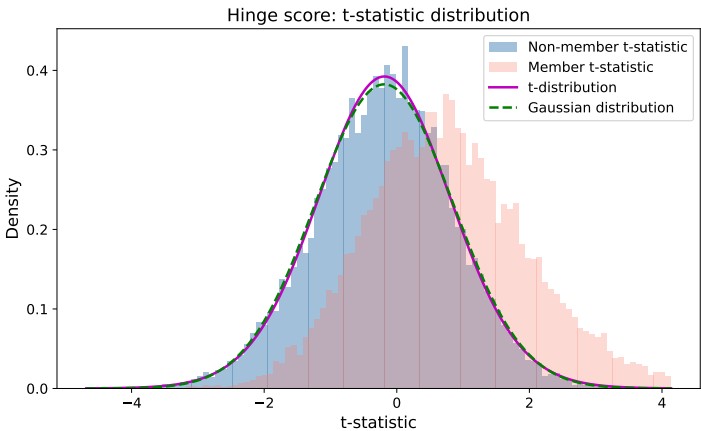

*Figure 6.* Empirical distribution of the $t$-test statistic constructed from hinge scores.

*Table 9.* Mean and standard deviation of attack performance (TPR (%) vs. FPR) on CIFAR-10. Results are obtained by applying LA to different reference models, each trained on a distinct subset of the data population.

| Statistic | TPR@FPR ↑ | | | |
|---|---|---|---|---|
| | 0.1% | 0.5% | 1% | 5% |
| mean | 2.84 | 6.65 | 9.48 | 21.65 |
| std | 0.58 | 0.64 | 0.63 | 0.89 |

### F.3. Log scale ROC curve.

Figure 7 displays the log scale ROC curves for CIFAR-10 models. In this visualization, BMIA achieves higher TPRs, particularly evident in the lower FPR regions.

## G. Details for Laplace Approximation

### G.1. Inference over Subsets of Weights

For the purpose of efficiency, we treat only the last layer weights probabilistically, that is the last-layer Laplace (Snoek et al., 2015; Kristiadi et al., 2020). Consider a neural network with $L$ layers, where the weight matrix of the last layer is $w^{(L)}$, and the parameters of the first $L-1$ layers are treated as a feature map. In the last-layer Laplace approach, the posterior distribution is approximated only over the weights of the last layer $w^{(L)}$, assuming that the parameters of the other layers are fixed at their MAP estimates. The posterior distribution for $w^{(L)}$ is then approximated as a Gaussian:

$$p(w^{(L)} \mid \mathcal{D}) \approx \mathcal{N}(w^{(L)} \mid w_{\text{MAP}}^{(L)}, \Sigma^{(L)}), \tag{43}$$

where $w_{\text{MAP}}^{(L)}$ is the MAP estimate for the last-layer weights, and $\Sigma^{(L)}$ is the covariance matrix of the approximation.

In the inference stage, we compute the features from the first $(L-1)$ layer, $h^{(L-1)}$, and then sample multiple values of $\mathbf{w}^{(L)}$ from this distribution. For each sampled $\mathbf{w}_i^{(L)}$, the output is computed as:

$$f_i = w_i^{(L)} h^{(L-1)}. \tag{44}$$

In this way, for each query, we perform the neural network inference only once.

### G.2. Complexity and Scalability

We next clarify the computational cost of the Laplace approximation used by BMIA. A full Laplace approximation over all model parameters is not practical for modern neural networks. If the model has $P$ trainable parameters, storing a dense

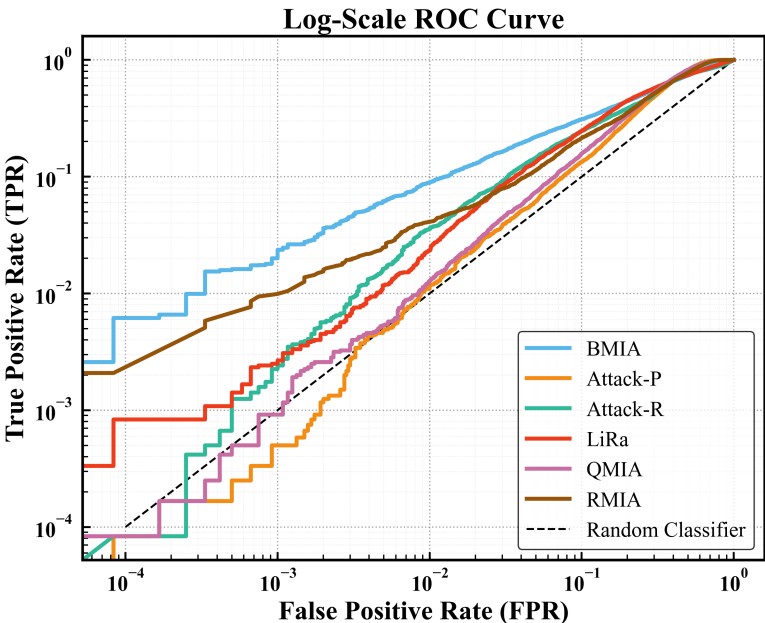

*Figure 7.* Log-scale ROC curve comparison of attacks on CIFAR-10. We use 2 reference models for Attack-R, LiRA, and BMIA.

Hessian or covariance matrix requires $O(P^2)$ memory, and exact construction or inversion is generally prohibitive. This is not the implementation used in our attack.

Instead, BMIA uses last-layer Laplace approximation. Let $d_{\text{in}}$ denote the feature dimension of the penultimate layer and let $C$ be the number of classes. The probabilistic parameter block is only the final linear classifier, whose size is $p_L = O(d_{\text{in}}C)$, independent of the number of parameters in the backbone. Therefore, increasing the depth or width of the feature extractor does not directly increase the Hessian storage, except through its effect on $d_{\text{in}}$.

The exact last-layer Hessian would require $O(p_L^2)$ storage. In practice, we use structured approximations. A diagonal approximation stores only $O(p_L)$ entries. With KFAC, the last-layer curvature is represented by two Kronecker factors of sizes $d_{\text{in}} \times d_{\text{in}}$ and $C \times C$, giving $O(d_{\text{in}}^2 + C^2)$ storage rather than $O(d_{\text{in}}^2 C^2)$. The cost of constructing these factors scales with the number of reference examples through standard forward and curvature accumulation, while posterior sampling and score computation for multiple samples are highly vectorized. For each query, the backbone feature is computed once, and the sampled last-layer classifiers are then applied to this fixed feature representation.

This complexity profile explains why the method is substantially cheaper than retraining many reference models while avoiding the infeasible full-Hessian regime. The remaining scalability cost comes mainly from the forward passes needed to compute features and accumulate the last-layer curvature, not from storing or inverting a Hessian over the entire model.

### G.3. Hessian Approximation

Consider the zero-mean Gaussian prior $p(w) = \mathcal{N}(w; 0, \gamma^2 I)$. The Hessian $\Sigma^{-1}$ is given by:

$$\nabla_w^2 \mathcal{L}(\mathcal{D}; w) = - \sum_{n=1}^{N} \nabla_w^2 \log p(y_n | x_n, w) \big|_{w=\hat{w}} + \gamma^2 I \tag{45}$$

In practice, the Hessian matrix $\Sigma^{-1}$ is approximated by the generalized Gaussian-Newton (GGN) matrix (Martens & Sutskever, 2011; Schraudolph, 2002; Martens, 2020):

$$G = \sum_{i=1}^{n} J(x_i) \left( \nabla_f^2 - \log p[y_i | f(x, w)] \big|_{w=\hat{w}} \right) J(x_i)^\top, \tag{46}$$

where $J(x_i) = \nabla_w f(x_i, w) \big|_{w=\hat{w}}$ is the Jacobian matrix of the model outputs w.r.t. $w$.

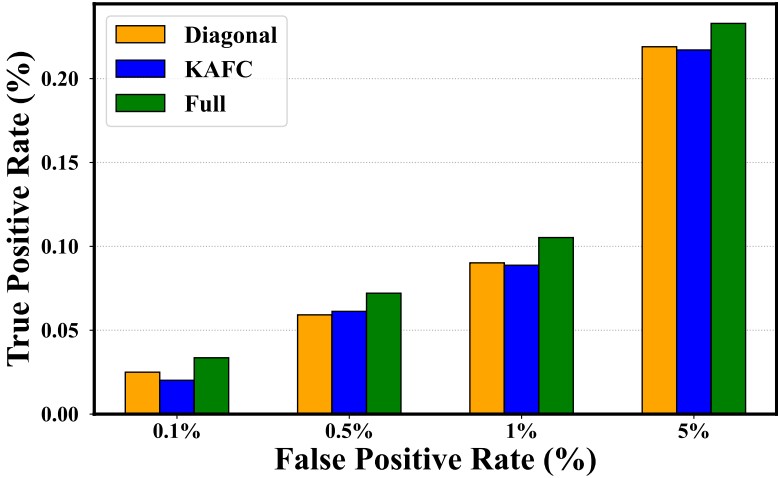

*Figure 8.* Attack performance comparison of variations of Hessian approximation on CIFAR-10.

However, $G$ is still quadratically large, we can further leverage diagonal factorization (LeCun et al., 1989; Denker & LeCun, 1990) and Kronecker-factored approximate curvature (KFAC) (Martens & Grosse, 2015; Heskes, 2000).

As shown in Figure 8, we evaluate the attack performance of BMIA using three distinct Hessian approximation methods. The KFAC approximation yields performance similar to the Diagonal method. Although the full Gauss-Newton method achieves the best performance, its higher inference time and storage requirements lead us to choose KFAC as the default method in this paper.

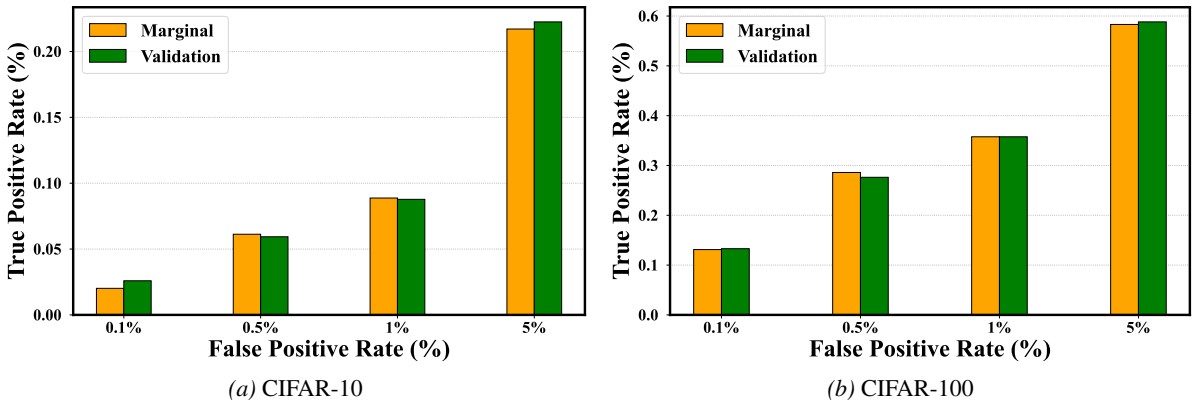

*(a)* CIFAR-10                                         *(b)* CIFAR-100

*Figure 9.* Attack performance comparison of two hyperparameter tuning methods on CIFAR-10/100.

### G.4. Hyperparameter Tuning

In LA, the prior $p(w)$ is set to be a zero-mean Gaussian $\mathcal{N}(w; 0, \gamma^2 I)$. In this paper, we mainly tune the variance hyperparameter $\gamma$ by maximizing the marginal likelihood (MacKay, 1992):

$$Z \approx \exp\left(-\mathcal{L}(\mathcal{D}; w_{\text{MAP}})\right) (2\pi)^{d/2} \left(\det \Sigma\right)^{1/2}.$$

In practice, $\gamma$ is tuned using gradient descent and does not require a validation set (Immer et al., 2021a). Alternatively, we can tune $\gamma$ by maximizing the posterior predictive over a validation set $\mathcal{D}_{\text{val}}$ (Ritter et al., 2018):

$$\gamma_*^2 = \arg\max_{\gamma^2} \sum_{n=1}^{N_{\text{val}}} \log p(y_n \mid x_n, \mathcal{D}).$$

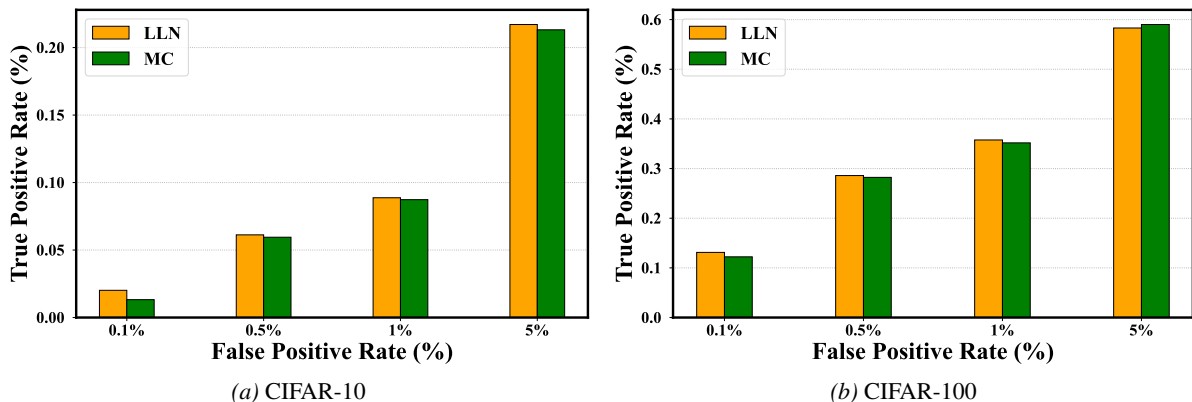

*(a)* CIFAR-10          *(b)* CIFAR-100

*Figure 10.* Attack performance comparison of two predictive distribution methods on CIFAR-10/100.

In Figure 9, we present the attack performance of BMIA using either the marginal likelihood or a validation set to tune $\gamma$. Empirically, these two methods do not show a significant difference. Therefore, we tune the prior hyperparameter by minimizing the marginal likelihood, as it does not require a validation dataset.

### G.5. Approximate Predictive Distribution

To obtain the score distribution for a given test query, the standard approach is Monte Carlo (MC) sampling. As shown in Algorithm 1, we sample weights $\tilde{\mathbf{w}}_i$ from the posterior $p(\mathbf{w} \mid \mathcal{D})$, and then compute the score samples $s_i = s(x^*, y^*; \tilde{\mathbf{w}}_i)$.

An alternative approach is to approximate the marginal distribution $f(x^*)$ for linearized neural network (LLN)(Khan et al., 2019; Foong et al., 2019; Immer et al., 2021b).

$$p(f^*|x^*, \mathcal{D}) = \mathcal{N}(f^*; f(x^*, \hat{w}), J(x^*)^\top \Sigma J(x^*)) \tag{47}$$

Subsequently, we sample the logits from this distribution and derive the score samples.

In Figure 10, we present the attack performance of BMIA with MC and LLN on CIFAR-10/100. Generally, LLN tends to perform slightly better than MC, but it may require more GPU memory because of the Jacobian matrix computation. Hence, we use MC for ImageNet-1k and LLN for other datasets.

