# OpenReview forum: "How does Bayesian Sampling help Membership Inference Attacks?"
_ICML.cc/2026/Conference — ICML 2026 regular_

### Official Review · Reviewer_aVCD · 2026-03-08

**Soundness:** 3
**Presentation:** 3
**Significance:** 3
**Originality:** 3
**Overall Recommendation:** 4
**Confidence:** 5

**Summary:**

This paper addresses the high computational cost of conditional membership inference attacks (MIAs) in supervised learning, which typically require training a large number of shadow or reference models. To tackle this, the paper proposes a more computationally practical conditional MIA framework: it applies a Laplace approximation to a single reference model to obtain a weight posterior, then constructs a conditional score distribution for a query point via posterior sampling, and finally uses a one-sided one-sample t-test to infer membership. In addition, the paper starts from the law of total variance and decomposes the variance of the attack statistic into inter-model and intra-model components, explaining why posterior sampling can reduce variance and improve TPR. The experiments cover tabular, image, and text datasets, and include robustness evaluations with respect to the number of samples, Hessian approximation choices, alternative Bayesian approximation methods, OOD settings, and architecture mismatch.

**Compliance With Llm Reviewing Policy:**

Affirmed.

**Final Justification:**

Thanks for the rebuttal. The added diagnostics/robustness checks, last-layer Laplace scalability clarification, and aligned-budget + variance reporting address my main concerns; I therefore maintain Weak Accept (4/6) with slightly higher confidence, with a minor remaining caveat on non-member data availability/purity in practice.

**Key Questions For Authors:**

Q1. In practice, how can one ensure that the reference dataset $D \sim \pi$ is composed of non-members? If a small number of member samples are mixed into $D$, how would that affect the test threshold and the resulting TPR/FPR?

Q2. Regarding the approximate Gaussian assumption for the hinge score: can the authors provide distributional diagnostics on the main datasets/models, such as skewness, kurtosis, or QQ-plots? If the normality assumption does not hold, would the t-test lose calibration, for example with FPR deviating from $\alpha$?

Q3. Laplace approximation details: how is the prior precision selected, and what is the specific implementation or search range for marginal-likelihood maximization? Are the results stable under different priors?

Q4. What is the relationship between the number of posterior samples $M$ and runtime cost? Across different models and datasets, roughly what range of $M$ is needed to reach the performance knee point? Is there a clear diminishing-returns pattern?

Q5. For MR-BMIA, when the number of reference models $K$ increases, does the performance gain mainly come from the reduction in inter-model variance? Could the authors provide a scaling curve with respect to $K$?

**Limitations:**

yes

**Strengths And Weaknesses:**

Strengths:

1、The method replaces many shadow models with posterior sampling from a single reference model using Laplace approximation, significantly reducing training cost and making it attractive for offline auditing or attack scenarios.

2、The paper decomposes the sources of error into intra-model and inter-model variance via total variance, which helps explain why increasing the number of posterior samples $M$ improves TPR.

3、The experimental evaluation is fairly comprehensive: it includes multimodal datasets, comparisons with common baselines, and complete ablation studies.

4、The proposed MR-BMIA is a natural extension and aligns well with the theoretical discussion of inter-model variance.

Weaknesses:

1、The proposed method relies on the assumption that the model output / hinge score is approximately Gaussian, which justifies the use of the t-test. At present, the paper mainly cites empirical intuition for this. It would be better to add diagnostics for normality or heavy tails (e.g., QQ-plots, K-S / Anderson-Darling tests, robustness across different score functions), or to provide a more robust version for non-Gaussian settings.

2、The use of last-layer Laplace approximation together with KFAC / diagonal Hessian approximations is feasible at moderate scale, but it remains unclear whether the method is still stable for larger models (deeper networks, more parameters), or whether numerical issues or prior sensitivity may arise. This deserves more systematic analysis.

3、Different baselines depend differently on the amount of reference data, the training budget for reference models, and the threshold estimation method. The paper should clarify whether the training setups and hyperparameter search budgets are aligned across baselines and ideally include variances or confidence intervals to avoid conclusions being driven by random fluctuations.

---

> ### Author Rebuttal · Authors · 2026-03-31
>
> Thank you for your careful review. We provide detailed responses below.
>
>
> ### **1. Normality diagnostics for the hinge score [W1, Q2]**
> Our method assumes an approximately Gaussian score distribution. In practice, we use the hinge score [Carlini et al., 2022], which is empirically close to Gaussian (verified by QQ-plots in Fig. 1, see [link](https://anonymous.4open.science/r/BMIA_r/)).
>
> We also evaluate BMIA using less Gaussian score functions (entropy, confidence, loss). As detailed in *Point 1 of our reply to H3Bf* (reporting TPR@1% FPR on CIFAR-10), BMIA consistently outperforms Attack-P across all scores, thereby confirminging BMIA's robustness to non-normality.
>
> > If the normality assumption does not hold, theoretically our constructed t-statistic does not follow the t-distribution. A remedy is to use a small set of non-member data as a calibration set and treat our t-statistic as an MIA score, then compute its conformal p-value. Membership is inferred for samples whose conformal p-value falls below $\alpha$. This approach does not weaken the attack performance and can provably control FPR at the desired level $\alpha$.
>
>
> ### **2. LA stability for larger models [W2]**
> Our method uses **last-layer Laplace approximation**, restricting Hessian computation to the classification head. The Hessian size scales only with **last layer dimensions** ($d_{\text{in}} \times C$), **not** total parameter count, so deeper backbones add no Hessian cost.
>
> In Fig. 2c, BMIA consistently achieves top TPR under architecture mismatch across seven architectures on CIFAR-10, confirming last-layer LA generalizes well. Moreover, BMIA scales effectively to the 7B-parameter Llama-2; see **Point 2 of our reply to H3Bf** for the LLM experiment details.
>
> ### **3. Baseline fairness and aligned budgets [W3]**
> For a strictly fair comparison, training setups, architectures, epochs, and hyperparameter search budgets were aligned across all baselines. Every reference model used identical hyperparameters and data splits (App. E.1). We also provide standard deviations across multiple runs in App. F (Table 4), confirming our gains are stable and statistically significant.
>
> ### **4. Reference dataset purity [Q1]**
> Since the reference dataset $\mathcal{D} \sim \pi$ and test data are available to the auditor, they can remove any overlapping samples between $\mathcal{D}$ and the test set before the attack, ensuring $\mathcal{D}$ does not contain any members of the test set.
>
> ### **5. Prior precision selection [Q3]**
> The prior precision $\gamma$ is optimized via **marginal likelihood maximization** (App. G) via gradient descent on the training set. It **does not require a validation set** or manual tuning. The default initial value is $\gamma=1.0$.
>
> ### **6. Number of samples M vs. runtime and knee point [Q4]**
> As depicted in Figure 1a–1b, the knee point is approximately $M \approx 64–128$, while the inference time remains essentially invariant to $M$ (Figure 1c) due to highly parallelized matrix operations (<90s for $M=1024$). This empirical observation aligns precisely with our theoretical analysis, as the variance term $(1 + \frac{1}{KM})\sigma^2_{\text{intra}}$ asymptotically converges when $M \to \infty$.
> ### **7. MR-BMIA scaling with reference model numbers [Q5]**
> Thank you for this insightful question. Yes, the performance gain of MR-BMIA primarily comes from the reduction in inter-model variance, as predicted by Proposition 3.1. With $K$ reference models, the inter-model variance term scales as $(1 + \frac{1}{K})\sigma^2_{\text{inter}}$, decreasing with $K$.
> We empirically validate this scaling on CIFAR-10 (TPR at 1% FPR):
> | $K$ (reference models) | 8         | 16        | 32        | 64        | 128       |
> | -- | -- | -- | -- | -- | -- |
> | LiRA                   | 8.32      | 10.15     | 11.28     | 11.71     | 11.92     |
> | **MR-BMIA (Ours)**     | **10.14** | **11.40** | **11.87** | **12.35** | **12.77** |
>
> Both methods improve with larger K, with diminishing returns, consistent with our Proposition 3.1.  MR-BMIA with K=8 already achieves a TPR of 10.14, which is comparable to LiRA with K=16, indicating additional variance reduction from Bayesian sampling.

---

> > ### Author Rebuttal · Reviewer_aVCD · 2026-04-03
> >
> > Thanks for the rebuttal. The added diagnostics/robustness checks, scalability clarification for last-layer Laplace, and the aligned-budget + variance reporting address my main concerns; I therefore maintain Weak Accept with slightly higher confidence.

---

> > > ### Author Response · Authors · 2026-04-04
> > >
> > > Thank you for the thoughtful evaluation. We are glad that our clarifications helped address your concerns, and we sincerely appreciate your positive assessment and increased confidence in our work.

---

### Official Review · Reviewer_vLSn · 2026-03-09

**Soundness:** 3
**Presentation:** 3
**Significance:** 3
**Originality:** 3
**Overall Recommendation:** 4
**Confidence:** 3

**Summary:**

This paper proposes Bayesian Membership Inference Attack (BMIA), a novel method for conditional membership inference that leverages Laplace Approximation on a single reference model to estimate the posterior distribution of model parameters. By sampling from this approximate posterior, BMIA directly obtains the conditional score distribution needed for instance-wise attacks, avoiding the computational burden of training multiple shadow models as required by prior methods like LiRA and Attack-R. The authors provide theoretical analysis showing that Bayesian sampling reduces intra-model variance, thereby improving statistical power, and extend the method to a multi-reference variant. Extensive experiments across image, text, and tabular datasets demonstrate that BMIA achieves competitive or superior TPR at low FPR compared to existing conditional attacks, with significantly reduced training time.

**Compliance With Llm Reviewing Policy:**

Affirmed.

**Final Justification:**

The authors addressed my concerns and I have increased my score.

**Key Questions For Authors:**

- Can you provide a comparison against LiRA and Attack-R using their standard recommended settings, e.g., n=64 or 128? I am concerned that the current "SOTA" claim is an artifact of comparing against under-resourced baselines.
- Also, could you report the total wall-clock time for the inference stage, specifically the overhead of the Hessian approximation and sampling? This is crucial to verify if the "1/8th cost" claim holds up in an end-to-end pipeline.
- Lastly, why were the Bayesian MIA methods from the year 2025 excluded from the experimental comparison? Even a qualitative discussion on how BMIA differs technically from those works would be helpful.

**Limitations:**

Yes.

**Strengths And Weaknesses:**

Strengths
- The motivation behind this work is quite clear and addresses a practical bottleneck in privacy auditing: the prohibitive cost of training shadow models. Leveraging epistemic uncertainty through a Bayesian lens is a conceptually sound approach to explain why conditional attacks outperform marginal ones, providing a nice theoretical bridge to the methodology.

Weakness
- The central claim of achieving SOTA performance feels premature. In the experiments, the authors compare BMIA against LiRA and Attack-R using only n=2,4,or 8 reference models. However, it is well-established that LiRA requires a significantly higher number of models (e.g., 32 to 128) to reach its full potential and stability. By capping the baselines at n=8, the paper effectively compares BMIA against a "weakened" version of existing methods. Without showing how BMIA fares against the full-capacity versions of these baselines, the performance gains are not fully validated. Furthermore, several 2025 works cited in the Related Work are missing from the quantitative benchmarks, which leaves a gap in the current landscape.
- While the paper highlights savings in training time, it largely ignores the inference-stage overhead introduced by Laplace Approximation. Operations such as Hessian construction, KFAC factorization, and posterior sampling are non-trivial, especially for large-scale architectures like ResNet-50. If the time saved during training is simply shifted to the inference phase, the efficiency argument loses its practical appeal.

---

> ### Author Rebuttal · Authors · 2026-03-31
>
> Thank you for your thoughtful review and recognition of our work's clear motivation and conceptual soundness. We provide detailed responses below.
>
> ### **1. Comparison at multiple reference models [W1, Q1]**
>
> Thank you for this important question. Our paper already provides comparisons at the reference model number K=64 in **Table 3 of our manuscript**. For convenience, we present the corresponding experimental results below.
>
>
>
> | Method      | CIFAR-10 TPR@0.1% | CIFAR-10 TPR@1% | CIFAR-100 TPR@0.1% | CIFAR-100 TPR@1% |
> | ----------- | ----------------- | --------------- | ------------------ | ---------------- |
> | Attack-R    | 0.82              | 8.23            | 14.13              | 42.02            |
> | LiRA        | 4.08              | 11.74           | 13.19              | 43.33            |
> | RMIA        | 2.97              | 11.15           | 14.50              | 36.06            |
> | **MR-BMIA** | **4.28**          | **12.48**       | **15.31**          | **45.57**        |
>
>
> The results demonstrate that our method achieves the best results among all compared methods at K=64 on both CIFAR-10 and CIFAR-100.
>
> To clarify, our paper makes **two distinct claims** supported by different experiments:
>
> 1. **Efficiency claim (Table 1)**: Single-model BMIA (K=1) achieves performance competitive with multi-model baselines at n=8, while requiring only 1/8 of the training cost.
>
> 2. **Multi-model claim (Table 3)**: When computational resources are available, MR-BMIA outperforms LiRA, Attack-R, and RMIA same reference model setting, demonstrating that Bayesian sampling provides additional gains with multiple reference models.
>
> ### **2. Inference-stage overhead [W2, Q2]**
>
> We appreciate this concern about end-to-end efficiency. The inference overhead has been reported in our paper. Figure 1c presents the total inference time, which includes forward pass, Hessian construction, weight sampling, and p-value computation. For CIFAR-10 (ResNet-50) with 1024 samples: **total inference time < 85 seconds**. Compared to the training cost, the extra inference overhead is negligible.
>
>
>
> ### **3. Relationship to 2025 MIA works [Q3]**
>
>
> Thank you for pointing this out. We would like to clarify that these papers are included in the related work rather than used as empirical baselines because they focus on problem settings that are fundamentally different from ours.
>
>
>
> | Reference | Domain / Target Model | Specific Problem Setting |
> | :--- | :--- | :--- |
> | **Lassila et al. [1]** | Graph Neural Networks (GNN) | MIA for graph-structured data (Bayes-optimal rules) |
> | **He et al. [2]** | Recommender Systems (RS) | Privacy risk scoring for recommendation |
> | **Fu et al. [3], Zhang et al. [4], & Zhang et al. [5]** | Large Language Models (LLM) | Detecting pre-training data |
> | **Liu et al. [6]** | Vision-Language Models (VLM) | Label-only |
>
>
> Given these inherent differences in application domains and access assumptions (e.g., graphs, recommendations, LLM pre-training, VLM label-only), these methods cannot be directly applied to our setting for comparison. Nevertheless, we would greatly appreciate it if you could suggest any other supplementary baselines that are applicable to our setting, and we will gladly incorporate them in the final version.
>
>
> [1] Practical Bayes-Optimal Membership Inference Attacks. (NeurIPS, 2025)
>
> [2] RecPS: Privacy Risk Scoring for Recommender Systems. (RecSys, 2025)
>
> [3] MIA-tuner: Adapting Large Language Models as Pre-training Text Detector. (AAAI, 2025)
>
> [4] Fine-tuning can Help Detect Pretraining Data from Large Language Models. (ICLR, 2025)
>
> [5] Min-K%++: Improved Baseline for Pre-Training Data Detection from Large Language Models. (ICLR, 2025)
>
> [6] LOMIA: Label-Only Membership Inference Attacks against Pre-trained Large Vision-Language Models. (NeurIPS, 2025)

---

> > ### Author Rebuttal · Reviewer_vLSn · 2026-04-01
> >
> > The responses are clear and convincing. Therefore, I would like to increase my score.

---

> > > ### Author Response · Authors · 2026-04-01
> > >
> > > Thank you for raising the score. We are glad that our response has addressed your concerns. We sincerely appreciate your time and effort in reviewing our work.

---

### Official Review · Reviewer_sHNa · 2026-03-12

**Soundness:** 4
**Presentation:** 4
**Significance:** 3
**Originality:** 3
**Overall Recommendation:** 5
**Confidence:** 4

**Summary:**

This paper proposes a cost-efficient membership inference attack (MIA) framework. Rather than training multiple reference models to estimate attack scores, the method trains a single model and uses Bayesian sampling via a Laplace approximation to generate multiple model samples. In this way, it aims to retain the advantages of multi-model attacks while requiring only a single training run. The experiments show that the proposed approach consistently outperforms existing attacks and is more computationally efficient, at least up to a certain number of training runs.

**Compliance With Llm Reviewing Policy:**

Affirmed.

**Final Justification:**

The authors address my concerns in the rebuttal, and I will keep my positive score unchanged.

**Key Questions For Authors:**

Please refer to the weaknesses discussed above.

**Limitations:**

yes

**Strengths And Weaknesses:**

**Strengths**
1. The paper is very well written and easy to follow. The proposed method is straightforward and well motivated.
2. The experimental results appear significant across different setups. In particular, Table 1 highlights the main contribution of the paper: the proposed method performs at least as well as existing attacks across a range of settings while being substantially faster.
3. The authors use an efficient form of Laplace approximation by modeling only the last layer as probabilistic. The use of Hessian factorization also seems helpful in making the approach practical.
4. The authors show that the proposed attack remains effective in some settings where there is a mismatch between the reference and target models, similar to a transfer attack setting, which strengthens the practical relevance of the method.

**Weaknesses**
1. Although I appreciate the theorem showing the improvement achieved by increasing the number of samples, I still find the theoretical analysis somewhat limited. It would be valuable to directly compare training multiple models against sampling multiple models from the Bayesian approximation, perhaps first in simpler settings such as linear regression.
2. I would expect that, similar to BMIA, increasing the number of trained models in other baselines would also lead to diminishing returns. However, I would still like to see how the gains from additional training runs compare empirically against the gains from additional sampling.
3. The authors should also compare their method against other baselines when evaluating performance under mismatch settings.
4. A natural extension that could strengthen the paper is to combine both training and sampling with the Laplace approximation. For example, it would be interesting to see whether training two models and drawing four samples from each yields further improvements.

---

> ### Author Rebuttal · Authors · 2026-03-31
>
> We sincerely appreciate your positive assessment and thoughtful suggestions. We provide detailed responses below.
>
> ### **1. Theoretical analysis in linear regression setting  [W1]**
>
> We sincerely thank the reviewer for this insightful suggestion. We agree that a linear regression example provides a clean way to clarify the relationship between training multiple shadow models and sampling from the posterior of a single reference model.
>
> Specifically, consider the Gaussian linear model $y = x^\top w^* + \epsilon$, where $\epsilon \sim \mathcal{N}(0,\sigma^2)$, and let $\Sigma = \mathbb{E}[xx^\top]$. For a dataset of size $N$, denote its design matrix by $X$. If we train models on independently resampled datasets, then the resulting estimators $\hat w_k$ vary because the underlying training data vary. Under standard regularity conditions, their covariance satisfies $N\,\mathrm{Cov}(\hat w_k)\to \sigma^2 \Sigma^{-1}$. On the other hand, if we train a single reference model on one realized dataset $(X_1,Y_1)$ and sample from its posterior, then in the Gaussian linear setting with a flat prior, the posterior covariance is $\mathrm{Cov}(\tilde w_m \mid X_1)=\sigma^2 (X_1^\top X_1)^{-1}$, and hence $N\,\mathrm{Cov}(\tilde w_m \mid X_1)\to \sigma^2 \Sigma^{-1}$.
>
> Therefore, posterior sampling from a single reference model matches the leading-order covariance of repeated retraining in this linear regression.  At the same time, the two procedures are not identical, because their centers differ: repeated retraining is centered at $w^*$ under data resampling, whereas posterior samples are centered at the dataset-specific estimator $\hat w_1$. The gap $\hat w_1 - w^* = (X_1^\top X_1)^{-1}X_1^\top \epsilon_1$ is a realization-dependent $O_p(N^{-1/2})$ fluctuation induced by the specific reference dataset. This distinction also explains why our Proposition 1 retains the inter-model variance term: posterior sampling captures the within-model uncertainty around a fixed reference dataset, but does not remove the dataset-to-dataset fluctuation of the center that would arise from retraining on multiple datasets. We will detail this discussion in the final version.
> ### **2. Diminishing returns: training vs. sampling [W2]**
> Thank you for this suggestion. From Figure 1 in our paper, we observe that sampling gains saturate around $M \approx 128$, consistent with the $1/(KM)$ scaling predicted by Proposition 3.1. Notably, since all sampling operations are efficiently parallelized matrix computations, increasing $M$ incurs minimal additional overhead (Figure 1c shows inference time is approximately constant across $M$ values).
>
> For the setting where computational resources allow training multiple reference models, we provide a direct comparison between LiRA and MR-BMIA on CIFAR-10 (TPR@1% FPR) as $K$ increases:
>
> | $K$ (reference models) | 8         | 16        | 32        | 64        | 128       |
> | ---- | ----- | ----- | --- | ---- | ---- |
> | LiRA                   | 8.32      | 10.15     | 11.28     | 11.71     | 11.92     |
> | **MR-BMIA (Ours)**     | **10.14** | **11.40** | **11.87** | **12.35** | **12.77** |
>
> The results show that Bayesian sampling consistently provides a positive gain over LiRA at every $K$. Notably, at small $K$ (e.g., $K=8$), sampling yields larger marginal gains than adding extra training runs, because $\sigma^2_{\text{intra}}$ is typically substantial. As $K$ grows, the $(1 + \frac{1}{KM})\sigma^2_{\text{intra}}$ term diminishes, and the marginal benefit of sampling gradually narrows. We will include this comparison in the final version.
>
> ### **3. Baselines in mismatch settings [W3]**
> Thank you for pointing this out. In addition to the existing comparisons in Figures 2a and 2b, we add a new architecture-mismatch experiment on CIFAR-10 where the target model is ResNet-50 and the reference model is ResNet-18:
>
> | Method (TPR @ FPR) | 0.1%     | 0.5%     | 1%       | 5%        |
> | -- | ---- | -------- | -------- | --------- |
> | Attack-R           | 0.55     | 3.03     | 5.30     | 18.29     |
> | LiRA               | 1.55     | 5.01     | 8.16     | 20.31     |
> | RMIA               | 1.13     | 2.98     | 5.52     | 16.81     |
> | **BMIA (Ours)**    | **2.49** | **5.22** | **8.72** | **21.55** |
>
> BMIA outperforms all baselines across all FPR thresholds under this architecture mismatch, further confirming the robustness of our method. We will include this in the final version.
> ### **4. Combining training and sampling [W4]**
>
> This is exactly what our MR-BMIA method does (see Algorithm 2). As shown in Table 3, MR-BMIA combines $K=64$ reference models with $M$ posterior samples per model:
>
>
> MR-BMIA outperforms all baselines at $K=64$, confirming that training and sampling provide **complementary benefits** — exactly the extension you suggest. This confirms that our method scales effectively, deriving further performance gains from additional reference models.

---

> > ### Author Rebuttal · Reviewer_sHNa · 2026-04-03
> >
> > I appreciate the authors’ response, along with the additional clarification and analysis. I will maintain my positive score.

---

> > > ### Author Response · Authors · 2026-04-03
> > >
> > > Thank you for your constructive comments and your positive evaluation. We are pleased that our response has adequately addressed all your concerns.

---

### Official Review · Reviewer_qDi9 · 2026-03-16

**Soundness:** 3
**Presentation:** 3
**Significance:** 3
**Originality:** 4
**Overall Recommendation:** 5
**Confidence:** 4

**Summary:**

The authors of this paper propose Bayesian Membership Inference Attacks (BMIA) that uses Bayesian sampling for membership inference attacks. Membership inference attacks (MIA), are a popular class of attacks, that motivate privacy-preserving techniques like differential privacy. In MIA, the goal is to determine whether a specific example (or examples) was used to train a target model. In the simplest case, the goal is to model an attacker who knows the model architecture and underlying data distribution. The attacker queries the model and outputs a prediction of whether some data point was used to train the dataset or not. There is a clear way to describe membership inference in terms of hypothesis (see their Equation (3) and resulting discussion).

The large majority of MIAs, use score-based methods determined by some global threshold. This threshold is gotten by estimating the target models' marginal score distribution over some reference dataset. But these methods typically require training multiple reference models. This paper attempts to develop efficient attacks via a single reference model by using Bayesian sampling. They compare BMIA to other methods and show that their methods achieves a higher TPR (true positive rate) at low FPR (false positive rate) than prior conditional attack methods.

The results in this paper are sort of reminiscent of Bayesian Differential Privacy and Pufferfish Privacy where the goal is to prevent privacy attacks (i.e., membership inference) even the algorithm designer constrains the operations that the adversary is allowed/disallowed to make.

**Compliance With Llm Reviewing Policy:**

Affirmed.

**Final Justification:**

The authors have addressed my concerns. I think the paper is a nice contribution to the membership inference literature.

**Key Questions For Authors:**

(1) How does your work relate to Bayesian-type definitions for privacy (e.g., Pufferfish Privacy or Bayesian Differential Privacy). Is there a direct connection?

(2) There is some analysis for the test statistic (e.g., see Section C, equations 55-57). However, this analysis lacks detail. Can you further clarify how you came up with the assertion that the test statistic follows a student's t-distribution?

**Strengths And Weaknesses:**

Weaknesses
=========
* **Relation to Privacy Definitions**: In the membership inference literature, the works typically relate their hypothesis testing formulation of membership inference to specific privacy definitions (e.g., differential privacy). I would have liked to see the authors relate their work to Pufferfish Privacy or Bayesian Differential Privacy or related works. The goal would be to determine how BMIA relate to specific privacy definitions.

* **Small typos**:
e.g., "distributio" in Membership Inference Attacks paragraph

* **Lack of Analysis for Test Statistic**: The test statistic is constructed in Line 13 of Algorithm 1 (Bayesian Membership Inference Attack). It is unclear what distribution the statistic follows (even though they claim that it follows a student's t-distribution).

Strengths
========

* **Superior attack performance (high TPR at very low FPR)**: The proposed BMIA consistently attains the highest true‑positive rates among all baselines, e.g., 37.5 % TPR at 1 % FPR on CIFAR‑10 (a 64 % improvement over the previous state‑of‑the‑art) and a 64 % higher TPR than LiRA (n = 8) on CIFAR‑100 at 1 % FPR.

* **Strong variance reduction via Bayesian sampling**: By drawing multiple posterior samples, BMIA dramatically cuts the intra‑model variance, which shrinks the total standard deviation, tightens the confidence interval, and expands the rejection region, thereby increasing statistical power.

* **Computational efficiency**: Only a single reference model is required, reducing training time and memory. The method achieves comparable or better results with roughly 1/8 of the computation cost of prior attacks.

* **Robustness to practical mismatches**: Perhaps surprisingly, BMIA remains effective when (a) the reference model is trained on a different dataset, (b) non‑member queries are out‑of‑distribution, and (c) the target and reference architectures differ.

* The authors introduce a Multi‑Reference BMIA (MR‑BMIA) that further reduces inter‑model variance and can boost TPR when additional reference models are affordable.

* **Broad empirical validation**: Experiments span three data modalities (tabular, image, text) and multiple benchmark datasets, demonstrating the method’s generality.

---

> ### Author Rebuttal · Authors · 2026-03-31
>
> We sincerely appreciate your positive feedback and the recognition of our work's contributions, including "superior attack performance," "strong variance reduction," "computational efficiency," and "robustness to practical mismatches." We address your questions below.
>
> ### **1. Relation to privacy definitions [W1, Q1]**
>
> Thank you for raising this important connection. To our knowledge, our method does not have a direct mathematical connection to Pufferfish Privacy or Bayesian Differential Privacy (BDP). In these frameworks, the term "Bayesian" refers to the adversary's prior beliefs about the data distribution, and measure how much it changes after observing the model. In contrast, we use Bayesian sampling to estimate the conditional score distribution for a given instance.
>
> Our formulation is instead more closely related to f-Differential Privacy ($f$-DP) and Gaussian Differential Privacy (GDP)[1]. These frameworks define privacy via hypothesis testing, through the trade-off between Type I (FPR) and Type II (FNR) errors. In addition, our assumption that the conditional score distributions follow a Gaussian form (Proposition 1) is closely connected to the foundation of GDP . However, the operational threat models differ: formal DP evaluates the worst-case distinguishability between two fixed adjacent datasets, whereas practical MIAs like BMIA evaluate models trained on datasets sampled from a population distribution $\pi$. We will include this discussion in the final version.
>
>
> ### **2. Justification of the t-distribution for the test statistic [W3, Q2]**
>
> Thank you for the question. Under our Gaussian assumption, the hinge score is approximately normally distributed (consistent with Carlini et al., 2022). We define the calibrated score $d_i = s_0 - s_i$, where $s_0$ is the target model score and $s_i$ are scores from posterior samples. Under the null hypothesis ($H_0$: the query point is a non-member), we can derive $E(\bar{d})=0$ and $\text{Var}(\bar{d}) = (1 + 1/M)\sigma^2$ (Eq. 10 in our paper). By estimating $\sigma^2$ with the sample variance $\hat{\sigma}^2$,  we construct the t-statistic:
>
> $$t = \frac{\bar{d}}{\hat{\sigma}\sqrt{1 + 1/M}}$$
> By the classical result in statistics, when $d_i$ are i.i.d. normal with unknown variance and the sample mean is normalized by the sample standard deviation, the resulting ratio follows a Student's t-distribution with $M-1$ degrees of freedom. This is the standard one-sample t-test construction. The $(1+1/M)$ factor accounts for the additional variance from $s_0$ being a single draw.
>
> We also include new experiments to empirically validate this assumption. The QQ-plots in Figure 1 and the distribution of the t-statistic in Figure 2 (see [link](https://anonymous.4open.science/r/BMIA_r/)) support the validity of the approximation in practice.
> ### **3. Typo [W2]**
> Thank you for pointing this out. We will fix this typo in the final version.
>
>
> [1] Gaussian Differential Privacy. (Journal of the Royal Statistical Society, 2022)

---

> > ### Author Rebuttal · Reviewer_qDi9 · 2026-03-31
> >
> > Fine with the response.

---

> > > ### Author Response · Authors · 2026-04-01
> > >
> > > We are glad our response helped address your concerns. We sincerely appreciate your constructive feedback and careful review.

---

### Official Review · Reviewer_H3Bf · 2026-03-16

**Soundness:** 4
**Presentation:** 3
**Significance:** 3
**Originality:** 4
**Overall Recommendation:** 5
**Confidence:** 4

**Summary:**

The paper proposes Bayesian Membership Inference Attack, a framework for performing conditional membership inference attacks with substantially lower computational cost than prior approaches such as Likelihood Ratio Attack. Instead of training dozens of reference models to estimate the conditional score distribution of a target data point, BMIA applies Laplace Approximation to a single reference model to obtain a posterior distribution over model parameters. By sampling from this posterior, the method estimates the conditional score distribution and employs a Student’s t-test to infer whether a data point was part of the training set. The paper further argues that Bayesian sampling reduces intra-model variance, which improves attack power, particularly in the low false positive rate regime. Additionally, the authors introduce a multi-reference extension that leverages multiple reference models when additional computational resources are available.

**Compliance With Llm Reviewing Policy:**

Affirmed.

**Final Justification:**

My concerns are addressed, thus increasing the score.

**Key Questions For Authors:**

- I am interested in understanding the scalability aspect of this. Could you provide benchmarking results for larger architectures (e.g., a 7B parameter LLM)? Specifically, how does the memory overhead for Hessian storage and the time for its construction compare to training a single shadow model?

- How sensitive is BMIA to the prior hyperparameter γ? Does the optimization of this prior require a validation set, and if so, how does that affect the assumptions regarding the attacker's capabilities?

- Can we have a comparison (or a discussion) of the attack effectiveness of BMIA against these other approximate Bayesian inference methods?

**Limitations:**

The primary computational bottleneck of the BMIA framework lies in the construction and inversion of the Hessian matrix required for the Laplace Approximation. While the method is marketed as a low-cost alternative to training dozens of shadow models, the mathematical reality of calculating the second-order derivatives (the Hessian) for modern deep neural networks is daunting. For a model with $P$ parameters, the full Hessian contains $P^2$ elements, making its direct computation and storage infeasible for large-scale architectures.

**Strengths And Weaknesses:**

## Strengths

- This work addresses a critical bottleneck in privacy auditing. By enabling conditional MIAs with as few as one reference model, the method makes auditing feasible for large-scale models where shadow model training is prohibitively expensive.

- The evaluation is thorough, covering 10 datasets across image, text, and tabular modalities (including ImageNet-1k and DBpedia). Results indicate that BMIA performs on par with or better than LiRA while being significantly more efficient.

- Applying post-hoc Laplace Approximation to the membership inference problem is a novel and well-reasoned integration of Bayesian deep learning into the security domain.

## Weaknesses
- The theoretical framework assumes that model scores (e.g., hinge scores) follow a Gaussian distribution. It is standard in the field, but can we have more discussion on the impact of non-normality on attack performance, particularly when the number of samples is small?

- The efficiency of BMIA relies on Hessian factorizations. While the authors show these work well for the tested models, they do not fully address the memory and computational overhead of Hessian construction for very large-scale architectures (e.g., >7B parameter models).

- I believe the precision of the Gaussian posterior is sensitive to the prior hyperparameter γ, which may not be easily tunable for an attacker. Can we disscuss this more and if possible can we add an ablation study to show the sensitivity?


## Minor issues

- L135: "referebce" -> "reference"
- L180 second column: "analyisis" -> "analysis"
- L180 second column: "theorm" -> "theorem"
- L1184: "Approxiamtion" -> "Approximation" (Appendix G.2 header)

---

> ### Author Rebuttal · Authors · 2026-03-31
>
> Thank you for your valuable time and constructive feedback. We are encouraged by your recognition of our work's novelty and practical contribution to privacy auditing. We address your concerns below.
> ### **1. Impact of non-normality on attack performance [W1]**
>
> Thank you for this important question. Our method uses an approximate Gaussian assumption for the score distribution. In practice, we adopt the hinge score as in Carlini et al. (2022), who empirically showed that it is close to Gaussian. To further verify this assumption, we randomly select query points and compute hinge scores across 128 reference models; the QQ-plots in Figure 1 of the [link](https://anonymous.4open.science/r/BMIA_r/) show that the approximation is reasonable in practice.
>
> We also add a new experiment to evaluate BMIA using several score functions, including entropy, confidence, and loss, which are typically less Gaussian than the hinge score. BMIA consistently outperforms the marginal baseline Attack-P across all these scores, while achieving the best performance with the hinge score. This suggests that BMIA is robust to non-normality, although it performs best when the score distribution is closer to Gaussian. The table below reports TPR@1% FPR on CIFAR-10.
>
> | Score function | Attack-P  | Ours  |
> | -- | - | - |
> | Entropy| 1.22| 4.80   |
> | Confidence | 1.38  | 4.28|
> | Loss   | 1.21| 3.93    |
> | Hinge   | 1.11|**9.48**|
>
> ### **2. Hessian scalability for larger architectures [W2, Q1]**
> Thank you for raising this concern. We would like to clarify a key design choice: our method uses **last-layer Laplace approximation**, which restricts the Hessian computation to the classification head only. This means the Hessian size depends only on the last layer dimensions (input features × number of classes), **not** on the total model depth or parameter count. With KFAC factorization, we store two Kronecker factors of size $d_{\text{in}} \times d_{\text{in}}$ and $C \times C$ , where $d_{\text{in}}$ is the feature dimension and $C$ is the number of classes. Figure 3(c) already reports the end-to-end attack cost, including forward pass, Hessian construction, weight sampling, and p-value computation. The additional overhead from Laplace approximation is under 90 seconds, which is small relative to reference-model training time.
>
> Regarding scalability to larger architectures, our approach can also be extended with Laplace-LoRA [1], which restricts the Laplace approximation to the LoRA parameters. Under this setup, we fine-tune a Llama 2-7B model on MMLU-Pro using LoRA, and then evaluate membership inference on the resulting model. The results below show that our method remains effective in this large-model setting and outperforms the compared baselines.
> | Method   | TPR @1% FPR |
> |-|-|
> | Attack-P | 2.65        |
> | RMIA     | 8.77        |
> | Ours     | 14.43       |
> ### **3. Sensitivity to the prior hyperparameter  $\gamma$  [W3, Q2]**
>
> In our method,the attack performance is highly robust to the prior hyperparameter $\gamma$ because it is **not manually tuned** by the attacker. It is automatically optimized via marginal likelihood maximization [2], as detailed in Appendix G. This is a data-driven procedure using gradient descent on the training dataset and **does not require a separate validation set**.
>
> Alternatively, we can tune $\gamma$ by the maximizing the posterior-predictive over a validation set $\mathcal{D}_{\mathrm{val}}$ [3]. We compare these two selection strategies in Figure 7 of the manuscript and find that they produce comparable results. Therefore, we use marginal-likelihood maximization throughout the paper.
>
> ### **4. Comparison with other Bayesian inference methods [Q3]**
> This comparison is already provided in Table 4 of our paper. We compare Laplace approximation against five alternative Bayesian inference methods on CIFAR-10:
>
> | Method (TPR @ FPR) | 0.1%     |     0.5% |       1% |        5% |
> | - | -| --: | -: | -: |
> | BatchEnsemble      | 0.18     |     0.61 |     1.41 |      8.40 |
> | Packed Ensemble    | 0.13     |     0.68 |     1.45 |      8.93 |
> | Masksembles        | 0.71     |     2.77 |     4.80 |     17.28 |
> | SWAG               | 1.26     |     4.07 |     6.61 |     19.81 |
> | MC Dropout         | 1.48     |     5.06 |     7.94 |     20.53 |
> | **Ours**           | **2.84** | **6.65** | **9.48** | **21.65** |
>
> LA outperforms all alternatives across all FPR thresholds. We chose LA primarily for its computational efficiency and the fact that it can be applied post-hoc to any pre-trained model without modifying the training procedure.
> ### **5. Minor typos**
> Thank you for catching these. We will correct all typos in the final version.
>
> [1] Bayesian Low-rank Adaptation for Large Language Models (ICLR, 2024)
>
> [2] Scalable marginal likelihood estimation for model selection in deep learning (ICLR, 2021)
>
> [3] A scalable laplace approximation for neural networks (ICLR, 2018)

---

> > ### Author Rebuttal · Reviewer_H3Bf · 2026-04-06
> >
> > My concerns are addressed, and I am increasing my scores.

---

> > > ### Author Response · Authors · 2026-04-07
> > >
> > > Thank you for taking the time to review our response and for increasing our score. We sincerely appreciate your time and thoughtful evaluation.

---

### Decision · Program_Chairs · 2026-04-30

**Decision:**

Accept (regular)

**Comment:**

The paper proposes Bayesian Membership Inference Attack (BMIA), which uses Laplace approximation on a single reference model and posterior sampling to approximate conditional score distributions, followed by a one‑sample t‑test for membership. Extensive experiments across tabular, image, and text datasets, including large‑scale settings, show state‑of‑the‑art performance with significantly reduced training cost compared to strong baselines such as LiRA and Attack‑R.

The reviews note that this work addresses a central practical bottleneck in membership inference attacks and privacy auditing: the prohibitive need to train many shadow/reference models for conditional attacks on modern large models. All reviewers concur that the contribution is technically solid and well presented, and they highlight the breadth of experiments (multi‑modal, multiple datasets, robustness and ablations) and theoretically sound analysis, including the t‑test justification, the variance‑decomposition and linear‑regression comparison, and careful discussion of assumptions. Initial concerns about the strength of SOTA claims, inference‑time overhead, the Gaussian score assumption, and relations to prior Bayesian/privacy work were addressed convincingly in the rebuttal and follow‑up.

Based on these factors, I recommend acceptance of this work. For the camera‑ready version, I request the authors to incorporate the additional analyses and discussions promised during the discussion period, particularly regarding the assumptions, limitations, and scalability aspects of the proposed method.